# A DYNAMICAL VIEW OF THE QUESTION OF *Why*

**Mehdi Fatemi**[*,1] **and Sindhu Gowda**[*,2]
[1] Microsoft Research, [2] University of Toronto and Vector Institute
mehdi.fatemi@ieee.org    sindhu.gowda@mail.utoronto.ca

## ABSTRACT

We address causal reasoning in multivariate time series data generated by stochastic processes. Existing approaches are largely restricted to static settings, ignoring the continuity and emission of variations across time. In contrast, we propose a learning paradigm that directly establishes causation between *events* in the course of time. We present two key lemmas to compute causal contributions and frame them as reinforcement learning problems. Our approach offers formal and computational tools for uncovering and quantifying causal relationships in diffusion processes, subsuming various important settings such as discrete-time Markov decision processes. Finally, in fairly intricate experiments and through sheer learning, our framework reveals and quantifies causal links, which otherwise seem inexplicable.

## 1  INTRODUCTION

Philosophers have long dreamed of discovering causal relationships from raw data. There are a wide variety of theories of causation, relevant to our discussion are the counterfactual theory (Lewis, 1973; 1979; 1986; 2000) and process-based theory of causation (Salmon, 1984; Dowe, 2000). The basic idea of counterfactual theories of causation is that the meaning of causal claims can be explained in terms of counterfactual conditionals of the form "If cause event $A$ had not occurred, effect event $B$ would not have occurred". The original counterfactual analysis of causation, most widely discussed in the philosophical literature, is provided by David Lewis (Lewis, 1979; 2000). Lewis's stated probability of causation between events as follows: "The effect event $B$ depends probabilistically on a cause event $A$ if and only if, given $A$, there is a chance $x$ of $B$'s occurring, and if $A$ were not to occur, there would be a chance $y$ of $B$'s occurring, where $x$ is much greater than $y$."

Works such as the Causal Bayes Net (Spirtes et al., 2000) or Structural Causal Model (Pearl et al., 2000) explored a counterfactual approach to causation that employs the structural equations framework to answer the causal question using interventionist/manipulationist approaches to find counterfactuals (HP setting, Halpern and Pearl (2001; 2005)). However, these approaches can have severe limitations, especially when applied to dynamical systems. Interventions are often infeasible in physical systems (Cartwright, 2007), and one can never observe counterfactuals nor assess empirically the validity of any modeling assumptions made about them, even though one's conclusions may be sensitive to these assumptions (Dawid, 2000; Berzuini et al., 2012). Moreover, these approaches ignore the dynamics as well as the possibility of other interventions between events (Dawid, 2000). Further, these frameworks assume knowledge of causal dependencies or structural information between various events in the system. Constructing detailed structural models can be hard, even for domain experts (Spirtes, 2001).

Other philosophers have proposed an alternative conception of causality, featuring physical systems as causal processes (Salmon, 1984; Fair, 1979; Kistler, 2007). The cornerstone of Salmon's theory of causality is the notion of a causal process, defined as a spatiotemporal continuous entity having the capacity to transmit "information, structure and causal influence" (Salmon, 1984). He believed that processes are responsible for causal propagation, and provide the links connecting causes to effects. While this understanding of causation is meaningful on an abstract level, philosophers have argued that Salmon's causal mechanical explanation was too weak, because it envisaged a geometrical network of processes and interactions (transmission of marks (Salmon, 1984) or conserved quantities (Dowe, 2000)) but did not convey as to what properties should be taken as explanatory (Hitchcock, 1995). Further, the scenarios described by everyday and scientific causal claims (e.g. 'smoking causes

---

*Equal contribution. Parts of this work were done during the authors' affiliation with Microsoft Research.

lung cancer') are often rather complex such that the possibility of decomposing them into sets of individual interactions is clearly out of sight (Fazekas et al., 2021).

As a different example, consider playing a seemingly simple Atari game where losing a point prompts the question: what caused this outcome? In its most basic form, an Atari game encompasses nearly 30,000 variables *at each time step*, resulting in tens of millions of variables during a short gameplay, each assuming 256 discrete values. Beyond its staggering size, constructing a causal graph demands substantial domain knowledge to decipher the combinatorially larger number of graph connections. Moreover, interventions in an active game necessitate delving into the internal game engine to mechanically adjust state variables—an impossible operation. This dynamic causal problem mirrors challenges found in diverse systems, such as pinpointing the reasons behind a patient's stroke in an ICU, understanding the cause of a nuclear reactor malfunction, or elucidating why a particular protein ceases development. An open question is how to uncover causal links in complex dynamical settings with no graph, no human-level knowledge beyond data, and no need for impossible interventions.

Inspired by the above theories of causality, our approach seeks to establish/validate causal assertions through the examination of underlying dynamics, placing a strong emphasis on spatiotemporal, system-level thinking. In a physical process, if events are seen as changes of state or action variables, we can naturally answer causal questions originating from the emission of changes in the state-space, across time. To this end, (i) we begin by defining causation from a process-based viewpoint. (ii) We then present two fundamental lemmas, which enable us to: (A) construct two reinforcement learning problems, whose optimal value functions yield core metrics to understand causation, and (B) isolate and quantitatively assess the individual contributions of each state or action component to the causal metrics. These lemmas reframe the notion of causation as a machine learning problem, making it amenable to analysis using raw observational data. (iii) We examine our methodology through a series of complex experiments[1]. We present a detailed account of related works in Appendix A.

## 2 BASICS AND PROBLEM FORMULATION

We adopt Kalman's definition of *state*: the smallest collection of numbers which must be specified at time $t = t_0$ to enable predicting the system's behavior for any time $t > t_0$. Any dynamical system can be described from the state viewpoint (Kalman, 1960). Formally, state is a $n$-dimensional vector-space and is either fully observable or reconstructable from observations. At any given time, each state component is a *random variable*, and the state vector's evolution across time forms a (stochastic) process. It is desirable to also include alterable inputs, i.e., *action* variables. The evolution of state is a function of both the intrinsic dynamics and the temporally selected (extrinsic) actions. We present our formal results for generic dynamical systems obeying (continuous) diffusion processes. We then derive our algorithmic machinery, which covers discrete cases and model-free settings.

**Diffusion Processes.** Assume a filtered probability space $(\Omega, \mathbb{F}, P)$. Let the state vector form a continuous-time random process $\mathbf{X}(t, \omega)$ over the mentioned probability space (we often suppress $\omega$ for brevity). The process $\mathbf{X}(t)$ is a diffusion if it possesses the strong Markov property and if its sample paths are continuous w.p. (with probability) one.

Many physical, biological and economic phenomena are either reasonably modeled or well-approximated by diffusion processes (Karlin and Taylor, 1981). Further, discrete-time Markov processes can be well-approximated by diffusion processes. Conversely, a diffusion process (continuous-time) can be discretized to make a discrete-time Markov process with arbitrary level of accuracy (for a formal discussion, see Karlin and Taylor (1981), pp. 168–169). As a result, we will readily extend our formal results to design *discrete-time algorithms*, which are of special importance in practice.

Let $\Delta_h \mathbf{X}(t) = \mathbf{X}(t + h) - \mathbf{X}(t)$ be the change of state $\mathbf{X}(t)$ over a time interval of length $h$. We assume that the following limits exists: $\lim_{h \downarrow 0} \frac{1}{h} \mathbb{E}[\Delta_h \mathbf{X}(t) \mid \mathbf{X}(t) = \mathbf{x}] = \boldsymbol{\mu}(\mathbf{x}, \mathbf{u}, t)$ and $\lim_{h \downarrow 0} \frac{1}{h} \mathbb{E}[\{\Delta_h \mathbf{X}(t)\}^2 \mid \mathbf{X}(t) = \mathbf{x}] = \boldsymbol{\sigma}(\mathbf{x}, \mathbf{u}, t)$, where $\boldsymbol{\mu}$ is a vector of size $n$ and $\boldsymbol{\sigma}$ is a matrix of size $n \times n$ (they are referred to as infinitesimal parameters), and $\mathbf{u} \in \mathcal{U} \subseteq \mathbb{R}^m$ is a $m$-dimensional

---

[1]In causal reasoning, two distinct (but related) classes of questions are intrinsically relevant: (1) a cause event is assumed and possible effects are in question – causal inference (*what* is the result of using a certain medication?). (2) An effect event is assumed and possible causes and the extent to which they contributed to the effect are in question (*why* did the Chernobyl reactor explode?). We primarily focus on the latter class; nevertheless, the present core concepts and technical results can readily be used for causal inference as well.

action. We further assume that both $\boldsymbol{\mu}$ and $\boldsymbol{\sigma}$ are continuous functions of their arguments, $\boldsymbol{\sigma}$ is positive definite, and all the higher moments are zero. The state evolution can therefore follow the following differential form (Stokey, 2009):

$$\mathrm{d}\mathbf{X}(t,\omega) = \boldsymbol{\mu}\big(\mathbf{X}(t,\omega), \mathbf{u}(t)\big)\mathrm{d}t + \boldsymbol{\sigma}\big(\mathbf{X}(t,\omega), \mathbf{u}(t)\big)\mathrm{d}\mathbf{W}(t,\omega), \qquad (1)$$

where $\mathbf{W}(t,\omega)$ denotes the vector of standard Brownian motions. We assume $\mathbf{u}$ to be deterministic, bounded, and follow $\dot{\mathbf{u}} \doteq \phi(t)$. We let $\boldsymbol{\mu}$ and $\boldsymbol{\sigma}$ be stationary, however, it is straight to extend to stochastic actions and/or non-stationary infinitesimals. Further, the time variable can be augmented to the state vector to simply accommodate for non-stationary cases. Placed with initial state distribution and *reward* function (and with an obvious abuse of terminology), we deem a Markov decision process (MDP) as a general term to refer to a (continuous-time) diffusion or a discrete-time Markov decision process. The MDP is formally defined as a tuple $M = (\mathcal{X}, \mathcal{U}, R, \mathcal{P}_0)$. $\mathcal{X}$ and $\mathcal{U}$ are sets of possible states and actions, $R : \mathcal{X} \mapsto \mathbb{R}$ is a scalar reward function, and $\mathcal{P}_0$ is the distribution of initial states. Let actions be selected according to some policy $\mathbf{u}(t) = \boldsymbol{\pi}(\mathbf{X}(t), t)$. Starting from $\mathbf{x}$, the random variable corresponding to the (undiscounted) accumulated future rewards is called *return*, and its expectation is called *value function*: $V^{\boldsymbol{\pi}}(\mathbf{x}) \doteq \mathbb{E}\{\int_t^T R(\mathbf{X}(t',\omega))\mathrm{d}t' | \mathbf{X}(t) = \mathbf{x}\}$ with the trajectory terminating at time $T > t$. Further, $V^*(\mathbf{x}) \doteq \max_{\boldsymbol{\pi}} V^{\boldsymbol{\pi}}(\mathbf{x})$ is called *optimal value* function. Finally, we say that $\mathbf{X}$ *admits* one or more known components $x_j$ at time $t$ iff $X_j(t) = x_j$.

**Process-based Causality.**   As mentioned earlier, we posit that causal relationships are based on temporal dynamics. Any causal relationship contains two events: cause (event $A$) and effect (event $B$). We argue that in all logical arguments on causation, the following axioms are true:

 i. Causality necessitates time: a causal relationship is realizable solely along the time axis.

 ii. Cause happens before effect and the relationship is unidirectional from cause to effect.

 iii. A causal relationship may imply neither necessity nor sufficiency.

These axioms set the ground for a natural view of causation. Notably, (i) requires that an event must be associated with a point or an interval in time; otherwise, no causal argument can possibly be made about that event being the cause or effect of any other event. In the HP settings of causation, the time dependency often becomes implicit in the arguments (e.g., in causal graphs), but it may be a source of confusion; hence, we seek a formulation that inherently includes time. Therefore, we formally define an *event* as a *change* of one or more state or action components during a homogeneous time interval. The components involved in an event are called *ruling variables*. The time interval is assumed to be short enough such that the dynamics can be considered as monotone. This assumption highlights the fact that an event cannot be a long-term incident relative to the rate of changes in the environment. This definition further enables us to consider changes in the same variable happening at different points in time as different events, which can be very helpful in practical cases of interest. Next, (i) and (ii) necessitate that "$A$ causes $B$" implies "$B$ cannot cause $A$"; This helps resolve the question of what constitutes the direction of the causal relation between two events. Furthermore, (iii) necessitates that, in general, a causal relationship requires probabilistic views and non-binary measures. For example, if "$A$ causes $B$" and if $A$ does not happen, then in general, one cannot conclude $B$ necessarily will not happen. By the same token, if $A$ happens, it may not necessarily imply $B$ will also happen. In other words, an event may partially contribute in the occurrence of another event in the future, although the case that $A$ is a necessary and/or sufficient cause for $B$ is a possibility. This further addresses the problem of pre-emption since cases of preemption show us that causes need not be necessary for their effects (Gallow, 2022).

The central idea behind Lewis definition is that causes, by themselves, increase the probability of their effects. In the presence of actions, the probability of a future event's occurrence is not well-defined. Considering arbitrary policies for action selection, one may devise different chains of events after $A$. Following each such policy incurs a different probability for event $B$'s occurrence. Remark that if $A$ causes $B$ then under the most pessimistic version of such chains of events, still $x$ must be greater than $y$ in Lewis's definition. Hence, we set $x$ to be the *minimum* probability of $B$'s happening.

We define *grit* of a future event $B$ at state $\mathbf{X}$, denoted by $\Gamma_B(\mathbf{X})$, as the *minimum probability* that $B$ occurs if current state is $\mathbf{X}$. As discussed, the minimum is taken over future courses of actions. Similarly, *reachability* of a future event $B$ is denoted by $\Lambda_B(\mathbf{X})$ and is defined as the *maximum probability* of $B$'s occurrence starting from $\mathbf{X}$. In discrete settings, it is helpful to extend the

definitions to starting from a given state and a given action (with an overload of notation): $\Gamma_B(\mathbf{X}, \mathbf{u})$ and $\Lambda_B(\mathbf{X}, \mathbf{u})$.

We further argue that if the net impact of each variable is known (all ruling and *non*-ruling ones), then there is *no need* for the designed "interventions," (modifying the history), as the role of intervention is to mechanically separate the impact of a variable from the collective impact. We, therefore, postulate the following definition of causation:

**Definition 1** (Causation). *In a stochastic process, $A$ is a cause of $B$ if and only if*

*C1. Time-wise, conclusion of $A$ happens at or before beginning of $B$;*

*C2. Expected grit of $B$ strictly increases from before to after $A$. Moreover, until $B$'s occurrence, it never becomes the same or smaller than its value at $A$'s beginning;*

*C3. The contribution of $A$'s ruling variables in the growth of $B$'s expected grit is strictly positive and is strictly larger in magnitude than that of non-ruling variables with negative impact.*

Remark that the non-ruling variables can have positive, zero, or negative impacts on the change of $B$'s grit. The second part of condition *C2* necessitates that a future event must not nullify the impact of a cause. Condition *C3* above requires that the contribution of $A$'s ruling variables must both be positive and overshadow the negative impact of non-ruling ones. It then follows that even in the absence of non-ruling variables with a positive impact, $B$'s grit still increases by $A$; hence, $A$ is a cause. Moreover, grit is a random variable due to non-ruling variables at the beginning of $A$. The *expected grit* asserts that causation must hold under the expected starting point. Of note, one can set forth a strong notion of causation by replacing *C3* to assert that the contribution of $A$'s ruling variables is strictly larger than that of *all* non-ruling variables. This notion helps to identify an event as a *dominant cause*. In any case, the yet-open question is how to compute individual contributions. In the next section, we will establish formal results to answer this question.

## 3 FUNDAMENTAL LEMMAS

We present two foundational lemmas. In a nutshell, the first lemma is a generalization of Lemma 2 in Fatemi et al. (2021), and it broadly states that grit and reachability can be computed by the optimal value functions corresponding to two easily constructed reward functions. This lemma establishes the learning of value functions (hence reinforcement learning) as the principal learning paradigm for dynamical causal problems. The second lemma decomposes expected change of grit and reachability to the contribution of state and action components, which inherently enables causal analysis. These lemmas are core to our theory in that they enable formal and computational reasoning about causality, which will be presented in the rest of this paper. All the proofs are deferred to Appendix B.

**Lemma 1** (**Value Lemma**). *Let $[T, T']$ be the duration of event $B$'s occurrence, and the state only admits $\mathbf{x}_B$ at $t = T'$ (all states that admit $\mathbf{x}_B$ are terminal). Define two MDPs $M_\Gamma$ and $M_\Lambda$ identical to $M$ with their rewards being zero if $B$ does not happen. Otherwise, $R_\Gamma(\mathbf{X}(t)) = R_\Lambda(\mathbf{X}(t)) = 0$ for $t < T$; $\int_T^{T'} R_\Gamma(\mathbf{X}(t))dt = -1$; and $\int_T^{T'} R_\Lambda(\mathbf{X}(t))dt = 1$. Let $V_\Gamma^*(\mathbf{x})$ and $V_\Lambda^*(\mathbf{x})$ denote the optimal value functions (undiscounted) of $M_\Gamma$ and $M_\Lambda$, respectively. The followings hold for all $\mathbf{X} \in \mathcal{X}$:*

*1. $\Gamma_B(\mathbf{X}) = -V_\Gamma^*(\mathbf{X})$*

*2. $\Lambda_B(\mathbf{X}) = V_\Lambda^*(\mathbf{X})$*

**Lemma 2** (**Decomposition Lemma**). *Fix a filtered probability space $(\Omega, \mathbb{F}, \mathcal{P})$. Let $\mathbf{X} = \mathbf{X}(t, \omega)$ be a diffusion process with stationary infinitesimal parameters $\boldsymbol{\mu} = \boldsymbol{\mu}(\mathbf{X}, \mathbf{u})$ and $\boldsymbol{\sigma} = \boldsymbol{\sigma}(\mathbf{X}, \mathbf{u})$. Let grit and reachability exist and be differentiable twice in state. Let $\boldsymbol{\sigma}_i(\mathbf{X}, \mathbf{u})$ denote the $i$-th row of the matrix $\boldsymbol{\sigma}(\mathbf{X}, \mathbf{u})$. Finally, let a fixed action $\mathbf{u}$ be applied from time $t_1$ to $t_2$ and the state admits occurance of event $A$ between $t_1$ and $t_2$. The expected change of grit, $\mathbb{E}[\Delta_A \Gamma_B] = \mathbb{E}[\Gamma_B(\mathbf{X}(t_2, \omega)) - \Gamma_B(\mathbf{X}(t_1, \omega))|A]$, is expressed by the following formula:*

$$\mathbb{E}[\Delta_A \Gamma_B] = \sum_{j=1}^n \mathbb{E}\{g_j | A\} + \sum_{j=1}^n \mathbb{E}\{\dot{g}_j | A\} + \sum_{j=1}^n \sum_{\substack{i=1 \\ i \neq j}}^n \mathbb{E}\{\ddot{g}_{j,i} | A\}, \tag{2}$$

$$g_j \doteq \int_{t_1}^{t_2} \mu_j(\boldsymbol{X}, \boldsymbol{u}) \cdot \frac{\partial \Gamma_B}{\partial x_j}(\boldsymbol{X}) dt \tag{3}$$

$$\dot{g}_j \doteq \frac{1}{2} \int_{t_1}^{t_2} \boldsymbol{\sigma}_j(\boldsymbol{X}, \boldsymbol{u}) \cdot \boldsymbol{\sigma}_j^T(\boldsymbol{X}, \boldsymbol{u}) \cdot \frac{\partial^2 \Gamma_B}{\partial x_j^2}(\boldsymbol{X}) dt \tag{4}$$

$$\ddot{g}_{i,j} \doteq \frac{1}{2} \int_{t_1}^{t_2} \boldsymbol{\sigma}_i(\boldsymbol{X}, \boldsymbol{u}) \cdot \boldsymbol{\sigma}_j^T(\boldsymbol{X}, \boldsymbol{u}) \cdot \frac{\partial^2 \Gamma_B}{\partial x_i \partial x_j}(\boldsymbol{X}) dt \tag{5}$$

*The same formulation holds for reachablity.*

If change of action variables is to be considered as an event, then **u** is allowed to change and a similar term is also required for actions. By assumption **u** is not a stochastic process; thus, it only adds a deterministic term. Let $\dot{\boldsymbol{u}}(t) \doteq \boldsymbol{\phi}(t)$ and $\phi_k$ be the $k$-th component of $\boldsymbol{\phi}$. We need to consider $\Gamma_B \doteq \Gamma_B(\boldsymbol{X}, \boldsymbol{u})$, and the additional term $\sum_{k=1}^{m} \mathbb{E}\{h_k|A\}$ will be added to equation 2 with

$$h_k \doteq \int_{t_1}^{t_2} \phi_k(t) \cdot \frac{\partial \Gamma_B}{\partial u_k}(\boldsymbol{X}, \boldsymbol{u}) \, \mathrm{d}t \tag{6}$$

Remark that **u** may take more complex forms or even be a stochiastic process. Then, other terms should also be added to decomposition lemma. Although such expansions are straightforward, we do not consider them here, since in practice, changes of **u** is often seen as extrinsic events.

Using fundamental lemmas, we next present certain basic properties for grit and reachibility:

**Proposition 1** (Unity Proposition). *If grit of an event $B$ is unity at some state $\boldsymbol{x}$, then w.p.1 it will remain at unity. Moreover, this occurs if and only if $B$ will happen w.p.1 from $\boldsymbol{x}$ regardless of future actions and stochasticity.*

**Proposition 2** (Null Proposition). *If reachablity of an event $B$ is zero at some state $\boldsymbol{x}$, then w.p.1 it will remain at zero. Moreover, this occurs if and only if $B$ will almost surely never happen, regardless of future actions and stochasticity.*

**Proposition 3.** *Let actions be selected according to a fixed policy $\boldsymbol{u} = \boldsymbol{\pi}(\boldsymbol{X})$ over a fixed time interval. The resultant expected changes in grit and reachability of a future event $B$ are bounded as follows: (1) $\min_{\boldsymbol{\pi}} \mathbb{E}[\Delta \Gamma_B] \leq 0$, and (2) $\mathbb{E}[\Delta \Lambda_B] \leq 0$ for all $\boldsymbol{\pi}$. Further, the equality in both statements holds if transitions are deterministic.*

The unity proposition states that *one* is the (only) sticky value for grit: once it is reached, grit will remain at one until $B$ is forcefully reached, irrespective of any *intrinsic* or *extrinsic* future event. We will use this important property for proving the sufficiency of a cause. The null proposition, enables to reason about rejection of a future event. We will use this proposition to establish necessity of a cause. The third proposition provides anticipation for the *expected change* of grit and reachability (i.e., on average). Of note, in practice, a *learned* value function is often used in place of $V^*$, which may violate such properties to various degrees depending on the level of approximation errors.

## 4 FORMAL ESTABLISHMENT OF CAUSATION

Let $\varphi_A(j) = \mathbb{E}\{g_j|A\} + \mathbb{E}\{\dot{g}_j|A\} + \sum_{\substack{i=1 \\ i \neq j}}^{n} \mathbb{E}\{\ddot{g}_{j,i}|A\}$ be the impact of component $j$ on event $B$'s grit during event $A$ (likewise for actions). Using decomposition lemma, we can directly state the definition of causation in a mathematical form, which we call *proposition of causation*:

**Proposition 4** (Causation). *Let $A$ occurs over the interval $[t_1, t_2]$ and $\mathcal{D}_A$ be the set of $A$'s ruling variables. $A$ is a cause of $B$ if and only if*

1. *$A$ happens before $B$*

2. *$\mathbb{E}\{\Delta_A(\Gamma_B)\} > 0$ and $\mathbb{E}\{\Gamma_B(\boldsymbol{X}(t))\} > \mathbb{E}\{\Gamma_B(\boldsymbol{X}(t_1))\}$ for all $t > t_2$*

3. *$\sum_{j \in \mathcal{D}_A} \varphi_A(j) > -\sum_{j \notin \mathcal{D}_A} \min(\varphi_A(j), 0)$*

Proposition 4 judges $A$ as a whole. If $A$ contains more than one ruling variable, i.e., $|\mathcal{D}_A| > 1$, a comparison of their individual contributions will help discover spurious or redundant variables inside $A$. This can prove useful in the context of causal discovery.

**Key Properties of Causation** Our proposition of causation induces various desired properties, we discuss a number of them herein. We, however, remark that no such statements as presented in this section are required and they are provided to grant certain plausibility to the theory. Nevertheless, the actual merit of our theory lends itself to its practical implications.

Without loss of generality, let event $B$ have only one ruling variable, $X_b$, and if $B$ occurs, it will be over the time interval $[T, T']$; hence, $T'$ is either terminal or no reward afterwards. Using value lemma, decomposition lemma, the definition of value functions, and the fact that the reward function of $B$ is only a function of $X_b$, i.e., $r_B(\mathbf{X}) = r_B(X_b)$, it follows that

$$\frac{\partial \Gamma_B(\mathbf{X}(t))}{\partial X_j(t)} = -\frac{\partial V_\Gamma^*(\mathbf{X}(t))}{\partial X_j(t)} = -\frac{\partial}{\partial X_j(t)} \mathbb{E} \int_{t^+}^\infty r_B(\mathbf{X}(t'))\mathrm{d}t' = -\frac{\partial}{\partial X_j(t)} \mathbb{E} \int_T^{T'} r_B(\mathbf{X}(t'))\mathrm{d}t'$$

$$= -\mathbb{E} \int_T^{T'} \frac{\partial r_B(X_b(t'))}{\partial X_j(t)}\mathrm{d}t' = -\mathbb{E} \int_T^{T'} \frac{\mathrm{d}r_B(X_b(t'))}{\mathrm{d}X_b(t')} \frac{\partial X_b(t')}{\partial X_j(t)}\mathrm{d}t' \tag{7}$$

Similar equations can be derived for the second derivatives. These derivatives of $\Gamma_B$ are still random variables due to $\mathbf{X}(t)$, as the expectation operators (from the definition of value functions) only affect stochasticity after $t$. If $B$ happens, the term $\mathrm{d}r_B(X_b(t'))/\mathrm{d}X_b(t')$ is nonzero (by construction) over $[T, T']$. Consequently, the driver terms are the derivatives (sensitivity) of $X_b$ at a future time $t'$ to the $j$-th state component at an earlier time $t$. If the sensitivity is zero, then the contribution of $j$ in change of grit will render null.

To shed more light on this, let us expand the first derivative. Remark that $\mathbf{X}(\cdot)$ is a diffusion; hence, there exists a sequence of $m \geq 0$ stopping times from $t$ to $t'$, such that $t \leq \tau_1 < \tau_2 < \cdots < \tau_m \leq t'$. The strong Markov property of $\mathbf{X}$ asserts that state components at each stopping time are conditionally independent of their values at any time prior to the preceding stopping time. We therefore write

$$\frac{\partial X_b(t')}{\partial X_j(t)} = \sum_{i_1 \in \mathcal{D}_1} \cdots \sum_{i_{m-1} \in \mathcal{D}_{m-1}} \sum_{i_m \in \mathcal{D}_m} \frac{\partial X_b(t')}{\partial X_{i_m}(\tau_m)} \frac{\partial X_{i_m}(\tau_m)}{\partial X_{i_{m-1}}(\tau_{m-1})} \cdots \frac{\partial X_{i_1}(\tau_1)}{\partial X_j(t)} \tag{8}$$

where $\mathcal{D}_k$ is the set of all state variables, which appear in the (stochastic) differential equation of $X_{i_{k+1}}$, with $\mathcal{D}_m$ corresponds to those of $X_b$. Equation equation 8 shows how a change in $X_j$ at $t$ propagates through other components across time until reaching $X_b$ at $t'$, thus causing $X_b$ to change in a certain way during event $B$. This may also be seen as a formal materialization of what philosophers refer to as "*chain of events* from $A$ to $B$" (Lewis, 1973; Paul, 1998). Plugging equation 8 into equation 7 and then into decomposition lemma, we see how this chain of events eventually changes the expected grit of $B$. Using these as well as previous results, we can prove various core properties:

i. **Efficiency:** The collective contribution of all components during any time interval is equal to $\mathbb{E} \, \Delta\Gamma_B$ over that interval.

ii. **Symmetry:** If two variables are symmetrical w.r.t. $X_b$ (i.e., having exactly the same impact on the dynamics of other variables, which ultimately reach $X_b$), then switching them does not impact $\partial_j \Gamma_B$. Furthermore, their contributions in $\Delta\Gamma_B$ will be exactly the same provided that their respective $\mu$ and $\sigma$ are the same during the given time interval.

iii. **Null event:** Contribution of $X_j$ in $\Delta\Gamma_B$ is zero *if and only if* at some stopping time through the propagation chain of equation 8, $\mathcal{D}_k$ is empty (meaning that there is no link between $X_j$ at $t$ and $X_b$ at $t'$). Such an event is called *null event* w.r.t. $B$.

iv. **Linearity:** Let $A_i$, $A_j$, and $A_{i,j}$ be three events with the ruling variables $X_i$, $X_j$ and $\{X_i, X_j\}$, respectively. Then, the contribution of $A_{i,j}$ in $\Delta\Gamma_B$ is sum of the contributions of $A_i$ and $A_j$.

**Correlations vs. Causation.** Wrongly identifying correlations as causal links is a core problem in formal reasoning. We show that our theory nullifies such links. Consider three consecutive and non-overlapping events $A$, $A'$, and $B$, which occur in this exact order and possess distinct ruling variables. Let $A$ be the cause of both $A'$ and $B$, and consider two cases: Case 1: $A'$ also causes $B$, and Case 2: $A'$ has nothing to do with $B$; however, they are still correlated due to having the same cause, i.e., $A$. Using equation 8, we observe that in Case 1, if both $A$ and $A'$ are causes of $B$, then all $\mathcal{D}_k$'s must be non-empty (otherwise they cannot be a cause due to the null-event property). As a result, in this case, change of grit will become non-zero, meaning that proposition of causation correctly asserts both $A$ and $A'$ as causes of $B$. In direct contrast, in Case 2, by assumption $A'$ is

a null event for $B$ and at some stopping time after the conclusion of $A'$, the propagation of ruling variables of $A'$ towards $X_b$ is terminated (i.e., propagation of $A$ toward $A'$ and toward $B$ happens through different collections of $\mathcal{D}_k$'s). Thus, equation 8 implies $\partial_j \Gamma_B = 0$ for $j \in \mathcal{D}_{A'}$; hence, proposition of causation will correctly reject $A'$ as a cause. This same logic can be used to address the problem of late-preemption that counterfactual theories have difficulty handling (Gallow, 2022).

**Sufficiency and Necessity of a Cause.**    There are two further results of practical importance, namely, sufficiency and necessity of a cause. A sufficient cause is one that forcefully makes the effect occur in finite time. According to unity proposition, the necessary and sufficient condition that $B$ forcefully happens from state $\mathbf{x}$ is $\mathbb{E}\,\Gamma_B(\mathbf{x}) = 1$. Using this result, the following proposition is immediate:

**Proposition 5** (Sufficient Causation). *Let $X^+$ be the state at $A$'s conclusion. Then, $A$ is a sufficient cause for $B$ if and only if $A$ is a cause for $B$ (proposition of causation holds) and $\mathbb{E}\,\Gamma_B(X^+) = 1$.*

A necessary cause is an event without which the effect will never happen from the current state. Occurrence of a necessary cause does not guarantee the effect's happening, but it is required for the effect to happen. More formally, if a state $\mathbf{x}$ does not admit the occurrence of $A$, then $A$ is a necessary cause for $B$ from the state $\mathbf{x}$ if every trajectory from $\mathbf{x}$ to $B$ passes through $A$. That is, if $A$ is not reachable from $\mathbf{x}$, then so isn't $B$. Using null proposition, the following is therefore immediate:

**Proposition 6** (Necessary Causation). *Let $A$ be a unique event (i.e., ruling variables of $A$ admit certain values "only if" $A$ occurs) and let $X$ not admit conclusion of $A$. $A$ is a necessary cause at $X$ for the event $B$ if and only if $A$ is a cause for $B$, and $\mathbb{E}\,\Lambda_A(X) = 0 \implies \mathbb{E}\,\Lambda_B(X) = 0$.*

**Computational Machinery.**    We can approximate the integrals with summations of $M$ points using the trapezoidal rule ($M$ is a hyper-parameter). Let us use the first-order approximate of $\boldsymbol{\mu}$, which only depends on applying $\mathbf{u}$ at the $m$-th point, corresponding to the time $m \cdot (t_2 - t_1)/M$. To simplify the notation, define $\mathbf{X}(m) \doteq \mathbf{X}(m \cdot (t_2 - t_1)/M)$. Note that in discrete-time problems, we still need to interpolate these points between the actual time ticks of the environment. We use forward approximation of $\boldsymbol{\mu}$ at $m$ and backward approximation at $m+1$, which alleviates the need for triple-point data of action $\mathbf{u}$. This yields $\boldsymbol{\mu}_j(\mathbf{X}(m), \mathbf{u}) \simeq \boldsymbol{\mu}_j(\mathbf{X}(m+1), \mathbf{u}) \simeq (\mathbf{X}_j(m+1) - \mathbf{X}_j(m))/\Delta t$. This formula approximates $\boldsymbol{\mu}$ by the slope of the (hyper-)line segment between $\mathbf{X}(k)$ and $\mathbf{X}(k+1)$. Using one-step trapezoidal rule and $\Delta t = (t_2 - t_1)/M$, it therefore follows (note that $\Delta t$ cancels out): $g_j \simeq \frac{1}{2} \sum_{m=1}^{M} \left( \mathbf{X}_j(m+1) - \mathbf{X}_j(m) \right) \cdot \left( \partial_j \Gamma_B(\mathbf{X}(m)) + \partial_j \Gamma_B(\mathbf{X}(m+1)) \right)$. We call this equation *g-formula*. Similar formulas can be derived for $h$, $\dot{g}$, and $\ddot{g}$.

## 5    EXPERIMENTS

We present two illustrative examples that no existing method can tackle. Modeling dynamical systems as SCMs is computationally and memory intensive to a prohibitive degree, especially in systems with numerous variables (Koller and Friedman, 2009). Additionally, it typically requires causal discovery methods and domain expertise to establish causal graphs and system equations. In contrast, our method operates solely on raw observational data without accessing system equations or causal graphs. Furthermore, defining $\neg A$ events in interventionist frameworks is largely ambiguous, particularly in continuous spaces, and predicting intervention outcomes heavily relies on restrictive assumptions about interventions or system models (Peters et al., 2022; Hansen and Sokol, 2014). Consequently, existing methods are irrelevant for baseline comparisons.

**Atari Game of Pong.**    To understand causal reasoning in a real setting, we applied our theory to the Atari 2600 (Bellemare et al., 2013) game of Pong. Event $B$ is losing a score, and the question is why $B$ happens if the player does not move its paddle. For our study, we use the DQN architecture (Mnih et al., 2015), but set $\gamma = 1$ and $r = -1$ if losing a point (terminal state) and zero otherwise. The rest of hyper-parameters are similar to Mnih et al. (2015). We next use the Pytorch's *autograd* to compute the value function's gradient w.r.t. screen pixels, based on which we could compute g-formula with $M = 10$ computational micro-steps to compute the integral. Further details can be found in Appendix D. Illustrated in Fig. 1, the method accurately pinpoints not only the last steps where the paddle should have moved (49), but also the pixels corresponding to the ball's movement. From 48, at each step, the set of actions that can catch the ball increasingly shrinks and the expected grit of losing

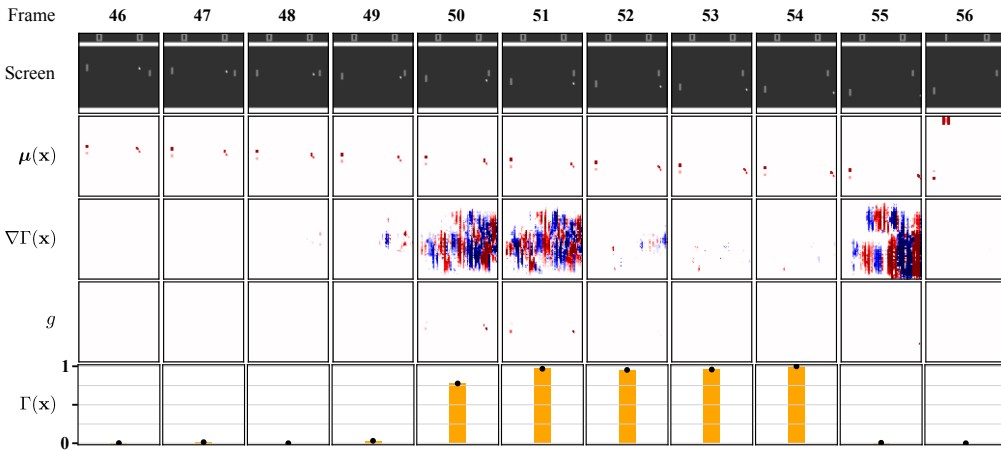

Figure 1: **Atari game of Pong.** $\nabla\Gamma$ is shown from red ($\geq 1$) to blue ($\leq -1$). Probing $g$ and $\Gamma(\mathbf{x})$ at frames 49 to 51 reveals the cause of losing score. Moreover, a *sufficient cause* is realized at frame 51.

a point increases. As all conditions in proposition of causation hold, these ball movements are the causes for $B$. Moreover, the change from 50 to 51 fulfills the proposition of sufficient causation. Playing the game step by step, one can easily confirm that 50 is the first frame, which is already too late and nothing can be done to catch the ball. Remark again that all these results are obtained through sheer learning with no access to system equations or human-level knowledge.

**Real-world Diabetes Simulator.** In this experiment, we analyze multivariate medical time-series data using an open-source implementation of the FDA-approved Type-1 Diabetes Mellitus Simulator (T1DMS) (Kovatchev et al., 2009). The simulator models a patient's blood glucose (BG) level and 12 other real-valued signals representing glucose and insulin values in different body compartments. We control two actions: 1) insulin intake, to regulate insulin level 2) meal intake, to manage the amount of carbohydrates. More details on the data and experiment design can be found in Appendix E. This experiment helps us investigate the impact of insulin and carbohydrate intake on blood glucose levels. Meal intake increases the amount of glucose in the bloodstream, which for T1D patients, is regulated with external insulin intake. However, intensive control of blood glucose with insulin injections can increase the risk of *hypoglycemia* (BG level < 70 mg/dL), which is the effect event we aim to understand its causes (event $B$). We use Monte Carlo learning to estimate $V_\Gamma(\mathbf{X})$ and thus $\Gamma_B(\mathbf{X})$. Of note, the algorithm has only access to data. In our setting, we observe that intake of insulin causes an instantaneous spike in dynamics of *subcutaneous insulin 1* (SI1), while intake of carbohydrates causes an instantaneous spike in *glucose in stomach 1* (GS1). Since the previous action is considered as part of the current state, the spike in subcutaneous insulin just acts as a proxy for action insulin, similarly for action meal and GS1. We study two scenarios under event $B$:

**(A) Single intake of insulin**: In the scenario described in Fig. 2-**A**, we wish to answer *why* event $B$ (BG level < 70 over $t \in [312, 313]$) did happen. Here we notice that $\Delta\Gamma_B(\mathbf{X}) > 0$ over $t \in [180, 181]$. Hence we can say that the change in state at $t \in [180, 181]$ contains possible causes for hypoglycemia. Now, we need to determine the change in which state component $X_i$ could have led to $\Delta\Gamma_B(\mathbf{X}) > 0$. To do this, we will decompose $\Delta\Gamma_B(\mathbf{X})$ as individual contributions from each state variable. Using decomposition lemma, in Fig. 3-**A** we see the individual contributions of each variable towards the total grit as seen in $g_i$. We notice that at the time interval $t \in [180, 181]$, the contribution to total grit only comes from variable SI1. Therefore, a change in variable SI1 at $t \in [180, 181]$ (event $A$) is considered cause of event $B$.

**(B) Multiple intakes of insulin and meal**: In Fig. 2-**B**, following a similar drill, we notice that $\Delta\Gamma_B(\mathbf{X}) > 0$ at $t \in [180, 181]$ and at $t \in [510, 511]$. However, although $\Delta\Gamma_B(\mathbf{X}) > 0$ at $t \in [180, 181]$, $\Gamma_B$ decreases to levels before $t=180$, prior to reaching event $B$. Therefore, change in variables at $t \in [180, 181]$ cannot be a cause since it violates condition C2 of proposition of causation. This only leaves change in variables at $t \in [510, 511]$ as a possible cause. Using decomposition lemma, Fig. 3-**B** reveals that at time interval $t \in [510, 511]$ the contribution to total grit comes only from variable SI1 (event $A_2$). Therefore event $A_2$ is the only cause of event $B$.

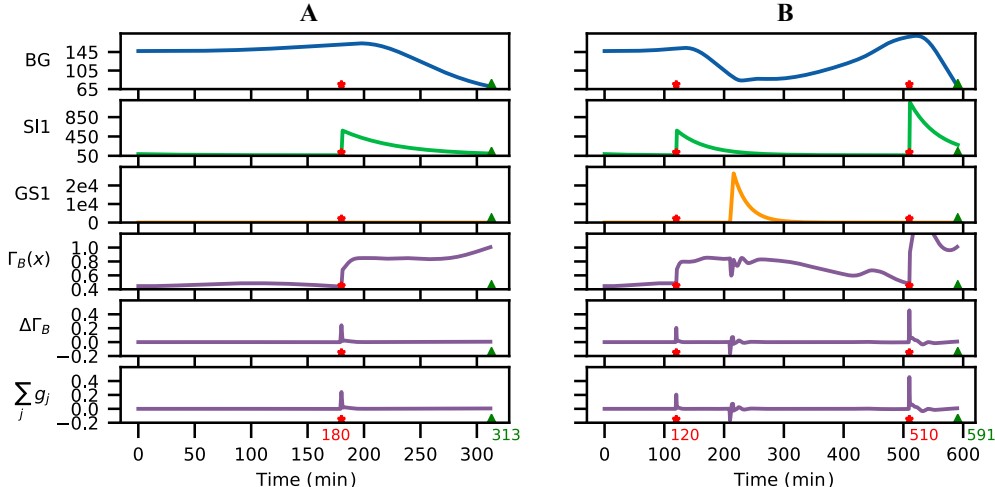

Figure 2: **Diabetes simulator.** Cause events are depicted with $*$ markers and the effect event $B$ (BG $< 70$) with ▲ marker. The simulation ends when event $B$ happens. The last two lines are change of grit computed from $V_\Gamma$ directly and from decomposition lemma (which are consistent).

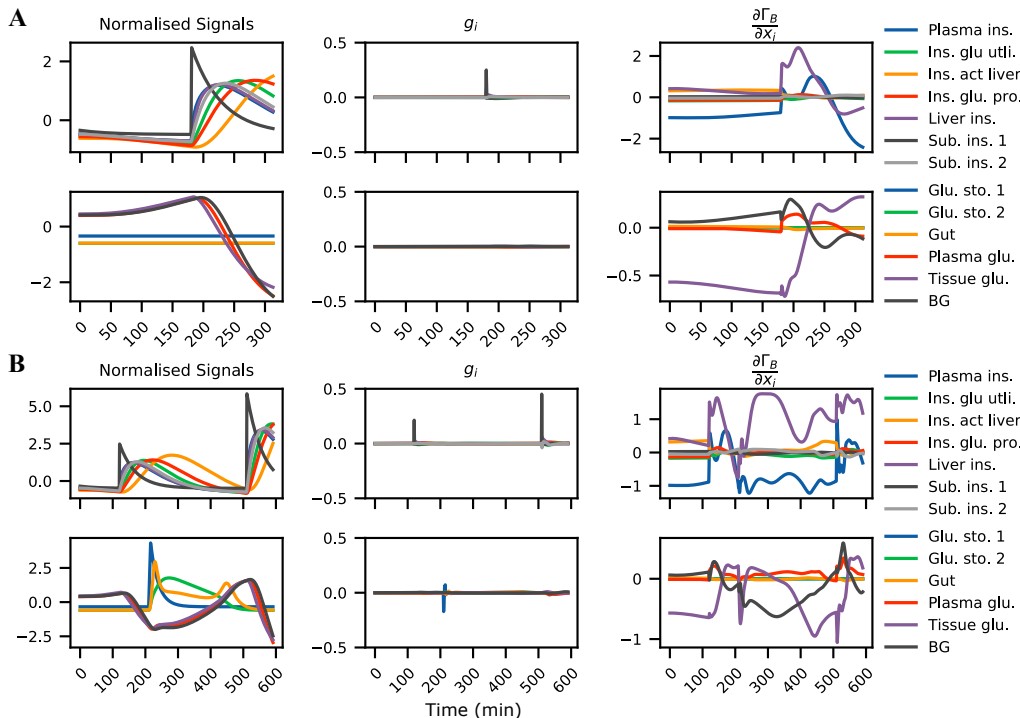

Figure 3: **Diabetes simulator.** Individual contributions of each signal towards the total grit.

## 6 CONCLUSIONS

We have presented a general theory for causation in dynamical settings, based on a direct analysis of the underlying process. We confined our exposition mostly to the case of given an event as effect, how to reason about possible causes. Our formal results enable a full framework, which can answer such causal questions directly from raw data. Further, we showed that various desired properties are immediate from our postulation, including the core conditions of counterfactual views. The main limitation of this work is two-fold: higher moments than two of the stochasticity are dismissed and the full information state is assumed. Relaxation of these assumptions are left for future work.

## DATA AND CODE AVAILABILITY

Our code and pretrained models to replicate the analysis (including figures) presented in this paper is publicly available at: `https://github.com/fatemi/dynamical-causality`.

For the T1D experiment we have used an open-source implementation of the FDA-approved Type-1 Diabetes Mellitus Simulator (Kovatchev et al., 2009). The code is publicaly available at: `https://github.com/jxx123/simglucose`

## ACKNOWLEDGMENTS

We express our sincere gratitude to our colleagues whose invaluable advice enhanced the quality of this work. Special thanks are extended to the former RL team and other colleagues at MSR Montreal for their pivotal suggestions during the early stages of this project. We are particularly grateful for the insightful feedback offered by Marzyeh Ghassemi, Elliot Creager, and Rahul G. Krishnan. Additionally, we would like to acknowledge the anonymous reviewers for their encouraging remarks and constructive suggestions, which helped to improve this paper.

Sindhu Gowda is supported by grants provided through Vector Institute and the University of Toronto. Special thanks is extended to her university advisor, Dr. Marzyeh Ghassemi, for all her supports and encouragements.

Resources used in performing this research were partially provided by Microsoft Research, the Province of Ontario, the Government of Canada through CIFAR, and companies sponsoring the Vector Institute: `www.vectorinstitute.ai/#partners`.

## AUTHOR CONTRIBUTIONS

This paper is the culmination of an extensive collaborative effort. MF initiated, conceptualized, and led the project. MF designed and developed theoretical concepts, formulated proofs, and implemented the Atari experiment. SG contributed significantly to discussions, integrating concepts and ideas from established causal literature, and providing diverse comparisons and discussions presented in both the main text and the Appendix. SG implemented the T1D experiment. Both authors jointly analyzed the results, co-authored the manuscript, and finalized the paper. The authors declare equal contributions to this work.

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

# A    RELATED WORK

**Static Settings.**    As noted above, philosophers have time and again proposed different theories trying to understand causation from raw data. Since the 80's, statisticians have tried to materialize this dream through mathematical reductions. The most popular framework for *actual causation* was purposed by Halpern and Pearl Halpern and Pearl (2001; 2005) following Pearl's influential book on causality that provided the first formal definition of causation Pearl et al. (2000). Pearl claimed that using causal models allows one to make the intuitive understanding of causation formally precise, whereas existing logical notions lack the resources to do so. Further, Pearl defined three basic probabilities of causation – the probability of necessity, of sufficiency, and of necessity and sufficiency, and ways to calculate them from data (Pearl, 2022). Moreover, researchers have used the causal structure and the properties of the data to narrow the bounds of the above probabilities of causation Tian and Pearl (2000); Dawid et al. (2017); Mueller et al. (2021). Needless to say, Pearl's account has come under criticism and revision – both from philosophers and researchers in AI Beckers (2021); Beckers and Vennekens (2018); Halpern (2015); Weslake (2015); Hitchcock (2001; 2007). However, all these works try to infer causal relationships from *non-temporal* data by making certain assumptions about the underlying process of data generation (causal graphs), which restricts the understanding of causation to static settings.

**Dynamic Settings.**    Works like Granger (1969); White and Lu (2010); Peters et al. (2013); Pfister et al. (2019); Eichler and Didelez (2012); Huang et al. (2020) deal with understanding causal relations in time series data, but mostly consider discrete-time models. Moreover, they focus on finding causal dependencies between different variables in time series data while we try to find causation between events, defined by a change in variables during a homogeneous time interval. Further, works like Peters et al. (2022); Hansen and Sokol (2014); Mooij et al. (2013); Blom and Mooij (2018); Bongers et al. (2018); Rubenstein et al. (2016) focus on continuous time systems that are governed by ordinary differential equations and propose a framework to model dynamical systems as structural causal models (SCMs). Again, they focus on understanding the effect of interventions and on causal structure learning or causal discovery under various system-level assumptions and do not deal with understanding the causation of events itself. It's important to note that "causal discovery" or structural learning as used in current literature deals with inferring the underlying causal structure or dependencies between variables from raw data Pearl (1980); Spirtes et al. (2000). It does not concern itself with understanding the cause of an event in a specific context Halpern (2016).

Further, all the above-mentioned methods deal with understanding causation from a counterfactual-interventionist perspective Woodward and Woodward (2005); Pearl (1980), while we follow the route of process theory of causation and emphasize system-level thinking to answer questions of causation Fazekas et al. (2021); Salmon (1984); Dowe (2000). Works like Fazekas et al. (2021) propose a philosophical framework for a dynamical systems approach to causation based on the process theory of causation Salmon (1984); Dowe (2000) and emphasize conceptually the importance of system-level thinking. However, the paper stops there, while we provide a formal framework that materializes this idea and enables computational machinery.

**Sensitivity Analysis.**    Another related area from a different domain is *sensitivity analysis*. The primary goal of sensitivity analysis is to understand the sensitivity or responsiveness of a model's output to variations in its input factors Ljung (1987); Skogestad and Postlethwaite (2005). However, it should be highlighted that sensitivity does not imply causation and defining causation purely based on sensitivity results in wrong causal arguments. Additionally, no connection is made in sensitivity analysis with value functions and learning algorithms thereof.

We believe a side-by-side discussion of dynamical systems and the theory of causation will allow us to develop novel approaches, transfer expertise across communities, and enable us to overcome the current limitations of each perspective individually. Our goal is to cast causation as a learning problem from dynamic temporal data such that given sufficient data, one can reliably answer the question of *why*. Notably, our paradigm conveniently covers both cases of intrinsic causation (cause is a change in the environment itself) and extrinsic causation (cause is an action applied to the environment).

## B    EXTENDED FORMAL RESULTS

Here, we present the proofs of formal claims from the paper with further discussions. The results are numbered as in the main paper.

**Lemma 1** (**Value Lemma**). *Define two MDPs $M_\Gamma$ and $M_\Lambda$ to be identical to $M$ with their corresponding reward kernels being $R_\Gamma = -\delta(\boldsymbol{x} - \tilde{\boldsymbol{x}})$ and $R_\Lambda = \delta(\boldsymbol{x} - \tilde{\boldsymbol{x}})$, where $\tilde{\boldsymbol{x}}$ admits the occurrence of event $B$ and $\delta(\cdot)$ denotes the Dirac delta function. Further, set all such $\tilde{\boldsymbol{x}}$ as terminal states. Let $V_\Gamma^*(\boldsymbol{x})$ and $V_\Lambda^*(\boldsymbol{x})$ denote the optimal value functions of $M_\Gamma$ and $M_\Lambda$, respectively, under $\gamma = 1$. Then, the followings hold for all $\boldsymbol{x} \in \mathcal{X}$:*

*1. $\Gamma_B(\boldsymbol{x}) = -V_\Gamma^*(\boldsymbol{x})$*

*2. $\Lambda_B(\boldsymbol{x}) = V_\Lambda^*(\boldsymbol{x})$*

*Proof.* For part 1, let $\pi^*$ denote an optimal policy of $M_\Gamma$. We note that since the only source of reward is when $B$ is reached and it is negative, then $\pi^*$ maximally avoids reaching $B$ (i.e., $\pi^*$ optimally chooses to reach anywhere but $B$). Hence, following $\pi^*$ results in the minimum probability of reaching $B$ from any state. On the other hand, $\gamma = 1$ induces that the return of any sample path is precisely $-1$ if $B$ is reached and zero otherwise. By definition, the optimal value of each state is the expectation of the return from all sample paths starting from that state and following $\pi^*$. Let $\mathcal{T}$ denote the set of all sample paths and partition it as $\mathcal{T} = \mathcal{T}_B \cup \mathcal{T}_N$, where $\mathcal{T}_B$ and $\mathcal{T}_N$ are disjoint sets corresponding to the paths which reach $B$ and those which do not (whose length can be of finite or infinite, the finite case occurs when there are terminal states in $M$ which may happen before ever reaching $B$ or when by assumption time horizon is finite). Let further $Z(\sigma)$ represent the return of a sample path $\sigma$ and $\mathcal{P}(\sigma|\mathbf{x}, \pi^*)$ denote the conditional probability that the sample path $\sigma$ occurs if $\pi^*$ is followed starting from the state $\mathbf{x}$. It then follows

$$V_\Gamma^*(\mathbf{x}) \doteq \mathbb{E}[Z \mid \mathbf{x}, \pi^*] = \int_{\sigma \in \mathcal{T}} Z(\sigma) \mathrm{d}\mathcal{P}(\sigma|\mathbf{x}, \pi^*)$$

$$= \int_{\sigma \in \mathcal{T}_B} -1 \, \mathrm{d}\mathcal{P}(\sigma|\mathbf{x}, \pi^*) + \int_{\sigma \in \mathcal{T}_N} 0 \, \mathrm{d}\mathcal{P}(\sigma|\mathbf{x}, \pi^*)$$

$$= -P_B^*(\mathbf{x})$$

where $P_B^*(\mathbf{x}) = \int_{\sigma \in \mathcal{T}_B} \mathrm{d}\mathcal{P}(\sigma|\mathbf{x}, \pi^*)$ denotes the total probability of reaching $B$ from $\mathbf{x}$ if $\pi^*$ is followed; that is, the minimum probability of reaching $B$ from $\mathbf{x}$, which by definition is $\Gamma_B(\mathbf{x})$.

Similarly, for part 2, let $\bar{\pi}^*$ denote an optimal policy of $M_\Lambda$. We write

$$V_\Lambda^*(\mathbf{x}) \doteq \int_{\sigma \in \mathcal{T}} Z(\sigma) \mathrm{d}\mathcal{P}(\sigma|\mathbf{x}, \bar{\pi}^*)$$

$$= \int_{\sigma \in \mathcal{T}_B} +1 \, \mathrm{d}\mathcal{P}(\sigma|\mathbf{x}, \bar{\pi}^*) + \int_{\sigma \in \mathcal{T}_N} 0 \, \mathrm{d}\mathcal{P}(\sigma|\mathbf{x}, \bar{\pi}^*)$$

$$= \bar{P}_B^*(\mathbf{x})$$

Here, $\bar{P}_B^*(\mathbf{x}) = \int_{\sigma \in \mathcal{T}_B} \mathrm{d}\mathcal{P}(\sigma|\mathbf{x}, \bar{\pi}^*)$ represents the probability of reaching $B$ from $\mathbf{x}$ if $\bar{\pi}^*$ is followed. $\bar{\pi}^*$ minimizes the chance of missing $B$ due to its positive reward and being the only source of reward. However, it only cares about reaching $B$ and it does not distinguish among various paths as long as they reach $B$. That is, $\bar{\pi}^*$ does not induce a shortest path to $B$, but it maximized the chance of reaching $B$. Hence, $\bar{P}_B^*(\mathbf{x})$ would be the maximum probability of reaching $B$ from $\mathbf{x}$, which by definition is $\Lambda_B(\mathbf{x})$, which completes the proof. We note that similar value functions, but for discrete time, state, and action, has also been introduced by (Fatemi et al., 2019; 2021).

$\square$

We should mention here that similar results may be extended to distributional RL (Bellemare et al., 2017) or to the case of semi-Markov settings (Sutton et al., 1999). Such settings are of practical interest (see for example Fatemi et al. (2022)).

**Lemma 2** (**Decomposition Lemma**). *Fix a filtered probability space $(\Omega, \mathbb{F}, \mathcal{P})$. Let $X = X(t, \omega)$ be a diffusion process with stationary infinitesimal parameters $\boldsymbol{\mu} = \boldsymbol{\mu}(\boldsymbol{x}, \boldsymbol{u})$ and $\boldsymbol{\sigma} = \boldsymbol{\sigma}(\boldsymbol{x}, \boldsymbol{u})$. Let grit and reachability be defined over $X$, and both be differentiable twice in state. Let $\boldsymbol{\sigma}_i(\boldsymbol{x}, \boldsymbol{u})$ denote the $i$-th row of the matrix $\boldsymbol{\sigma}(\boldsymbol{x}, \boldsymbol{u})$. Finally, let a fixed action $\boldsymbol{u}$ be applied from time $t_1$ to $t_2$ and the state admits certain values at $t_1$ and $t_2$ for some of its components. The admissions correspond to an event $A$. The expected change of grit, $\mathbb{E}\left[\Delta_A \Gamma_B\right] = \mathbb{E}\left[\Gamma_B\big(X(t_2, \omega)\big) - \Gamma_B(X(t_1, \omega))|A\right]$, is expressed by the following formula:*

$$\mathbb{E}\left[\Delta_A \Gamma_B\right] = \sum_{j=1}^n \mathbb{E}\left\{g_j | A\right\} + \sum_{j=1}^n \mathbb{E}\left\{\dot{g}_j | A\right\} + \sum_{j=1}^n \sum_{\substack{i=1 \\ i \neq j}}^n \mathbb{E}\left\{\ddot{g}_{j,i} | A\right\}, \tag{9}$$

$$g_j \doteq \int_{t_1}^{t_2} \mu_j(X, \boldsymbol{u}) \cdot \frac{\partial \Gamma_B}{\partial x_j}(X) dt \tag{10}$$

$$\dot{g}_j \doteq \frac{1}{2} \int_{t_1}^{t_2} \boldsymbol{\sigma}_j(X, \boldsymbol{u}) \cdot \boldsymbol{\sigma}_j^T(X, \boldsymbol{u}) \cdot \frac{\partial^2 \Gamma_B}{\partial x_j{}^2}(X) dt \tag{11}$$

$$\ddot{g}_{i,j} \doteq \frac{1}{2} \int_{t_1}^{t_2} \boldsymbol{\sigma}_i(X, \boldsymbol{u}) \cdot \boldsymbol{\sigma}_j^T(X, \boldsymbol{u}) \cdot \frac{\partial^2 \Gamma_B}{\partial x_i \partial x_j}(X) dt \tag{12}$$

*and the expectations are expressed on $\omega$. The same formulation holds for reachablity.*

*Proof.* Conditioning on event $A$ makes some components of $\mathbf{X}$ become deterministic and known, and the process is still a diffusion. Hence, the result follows from Itô's lemma, then taking conditional expectation from both sides and rearranging the terms. Remark that for any integrable function $\mathbf{Y}(t, \omega)$, we have $\mathbb{E}\{\int_{t_1}^{t_2} \mathbf{Y}(t, \omega) \, d\mathbf{W}(t, \omega)|A\} = 0$ (see Theorem 3.1 in Stokey (2009)). As a result, the $d\mathbf{W}$ part in Itô's lemma is eliminated and the stated result will follow.

$\square$

**Proposition 1** (Unity Proposition). *If grit of $B$ is unity at some state $\boldsymbol{x}$, then with probability one it will remain at unity. Moreover, this occurs if and only if $B$ will happen with probability one from $\boldsymbol{x}$ regardless of future actions and stochasticity.*

*Proof.* We establish the proof under mild assumptions on the dynamics (that a small enough $\Delta$ exists). A more rigorous proof may be possible by relaxing such assumptions (like it is in the discrete cases). However, insofar as the goal being applying the theory to practical problems, which naturally involve discrete or discretized time, the present proof fully suffices.

We first prove that if grit is unity then $B$ will happen w.p.1. and grit will remain at one until $B$ occurs. Following the value lemma, $\Gamma_B(\mathbf{x}) = -V_\Gamma^*(\mathbf{x})$. We therefore show that if $V_\Gamma^*(\mathbf{x}) = -1$, it will then remain at -1 until $B$ occurs. Remark that in the case of discrete state and discrete time, the result follows Lemma 1 of Fatemi et al. (2021) (similar ideas also exist in Fatemi et al. (2019) and Cao et al. (2023)). Here, using a similar line of argument, we present the proof for the general case of continuous time and state.

Remark that both $V_\Gamma^*$ and $Q_\Gamma^*$ are in $[-1, 0]$ for all states and actions. Thus, $-1 = V_\Gamma^*(\mathbf{x}) = \max_{\mathbf{u}} Q_\Gamma^*(\mathbf{x}, \mathbf{u})$ implies that $Q_\Gamma^*(\mathbf{x}, \mathbf{u}) = -1$ *for all* $\mathbf{u}$. Therefore, if $V_\Gamma^*(\mathbf{x})$ remains at -1, so does $Q_\Gamma^*(\mathbf{x}, \mathbf{u})$ for all $\mathbf{u}$ and, as a result, we only require to show $V_\Gamma^*(\mathbf{x})$ remains at -1 with no reference to any particular policy for action selection. In other words, all actions are optimal at $\mathbf{x}$ (w.r.t. maximizing integration of $R_\Gamma$) and choice of $\mathbf{u}$ at $\mathbf{x}$ makes no difference.

By construction, any trajectory that includes $B$ has a terminal state at the end of $B$. Let $\Delta$ be a small positive number such that it can cover the duration of $B$. We partition time into intervals of length $\Delta$. Starting from $\mathbf{x}$ at time 0, the world will be at a (random) state $\mathbf{X}'(\omega) \doteq \mathbf{X}(\Delta, \omega)$ at time $t = \Delta$. Let $\Delta$ be small enough such that selection of $\mathbf{u} \in \arg\max_{\mathbf{u}'} Q_\Gamma^*(\mathbf{x}, \mathbf{u}')$ at $t = 0$ and sticking to it for $[0, \Delta]$ is almost the same as following $\arg\max Q_\Gamma^*(\mathbf{x}, \cdot)$ during $[0, \Delta]$. Such $\Delta$ exists due to the continuity of diffusion's sample paths and the assumption that duration of any event, including $B$, has to be short w.r.t. the rate of changes of the state.

During the time interval $[0, \Delta]$, exactly one of four possibilities could occur:

1. a terminal state happens that admits $B$;

2. a terminal state happens that does not admit $B$;

3. no termination: $\mathbf{X}'$ is a non-terminal state with $V_\Gamma^*(\mathbf{X}') = -1$;

4. no termination: $\mathbf{X}'$ is a non-terminal state with $V_\Gamma^*(\mathbf{X}') = -\beta \in (-1, 0]$.

In the first two cases, by the definition of terminal state, $\mathbf{X}'$ is also a terminal state with zero value. Let $\mathcal{C}_1$ to $\mathcal{C}_4$ represent the sets of all possible states $\mathbf{X}'$ corresponding to each of the four cases above. These sets are mutually disjoint. We then show that if $V_\Gamma^*(\mathbf{x}) = -1$, then *only* either of (1) or (3) can happen. We note that any sample path of a diffusion is continuous. Since $R_\Gamma(\cdot)$ is also a continuous function, its integral exists. We can therefore write:

$$
\begin{aligned}
-1 = V_\Gamma^*(\mathbf{x}) &= \mathbb{E}\Big[ \int_0^\Delta R_\Gamma(\mathbf{X}(t, \omega))\mathrm{d}t + V_\Gamma^*(\mathbf{X}'(\omega)) \mid \mathbf{X}(0, \cdot) = \mathbf{x} \Big] \\
&= \mathcal{P}[\mathbf{X}' \in \mathcal{C}_1]\big( -1 + 0 \big) + \mathcal{P}[\mathbf{X}' \in \mathcal{C}_2]\big( 0 + 0 \big) + \mathcal{P}[\mathbf{X}' \in \mathcal{C}_3]\big( 0 - 1 \big) + \mathcal{P}[\mathbf{X}' \in \mathcal{C}_4]\big( 0 - \beta \big) \\
&= -\mathcal{P}[\mathbf{X}' \in \mathcal{C}_1 \cup \mathcal{C}_3] - \beta\mathcal{P}[\mathbf{X}' \in \mathcal{C}_4] \\
&= -\Big( 1 - \mathcal{P}[\mathbf{X}' \notin \mathcal{C}_1 \cup \mathcal{C}_3] \Big) - \beta\mathcal{P}[\mathbf{X}' \in \mathcal{C}_4] \\
&= -\Big( 1 - \mathcal{P}[\mathbf{X}' \in \mathcal{C}_2 \cup \mathcal{C}_4] \Big) - \beta\mathcal{P}[\mathbf{X}' \in \mathcal{C}_4] \\
&= -1 + \mathcal{P}[\mathbf{X}' \in \mathcal{C}_2] + \big( 1 - \beta \big)\mathcal{P}[\mathbf{X}' \in \mathcal{C}_4]
\end{aligned}
$$

which deduces

$$
\mathcal{P}[\mathbf{X}' \in \mathcal{C}_2] + \big( 1 - \beta \big)\mathcal{P}[\mathbf{X}' \in \mathcal{C}_4] = 0
$$

We remark that $1 - \beta$ is strictly positive; thus we conclude both $\mathcal{P}[\mathbf{X}' \in \mathcal{C}_2] = 0$ and $\mathcal{P}[\mathbf{X}' \in \mathcal{C}_4] = 0$. Consequently, the resultant state is either a terminal state admitting $B$ (i.e., $\mathbf{X}' \in \mathcal{C}_1$) or some state $\mathbf{X}'$ where $V_\Gamma^*(\mathbf{x}') = -1$ (i.e., $\mathbf{X}' \in \mathcal{C}_3$). Following the same line of argument on $\mathbf{X}'$ and noting that by assumption the time horizon is finite, we conclude that $V_\Gamma^*$ remains precisely at -1, and the path will eventually reach $B$ with probability one, regardless of stochasticity and selected actions.

Conversely, if from a state $\mathbf{x}$, event $B$ is going to happen with probability one, then all possible future trajectories will reach a reward of $-1$, which makes their return also be $-1$. More precisely, those trajectories which end with a reward of zero will have zero probability. Hence, the expected return from (i.e., the value function of) state $\mathbf{x}$ would be $-1$ regardless of stochasticity; hence, $V_\Gamma(\mathbf{x}) = -1$. It is then immediate from value lemma that $\Gamma_B(\mathbf{x}) = 1$ if $B$ occurs with probability one from $\mathbf{x}$.

$\square$

**Proposition 2** (Null Proposition). *If reachablity of an event $B$ is zero at some state $\mathbf{x}$, then w.p.1 it will remain at zero. Moreover, this occurs if and only if $B$ will almost surely never happen, regardless of future actions and stochasticity.*

*Proof.* The proof is similar to the previous proposition. In particular, during the time interval $[0, \Delta]$, exactly one of three possibilities could occur:

1. a terminal state happens that admits $B$;

2. a terminal state happens that does not admit $B$;

3. no termination: $\mathbf{X}'$ is a non-terminal state with $V_\Lambda^*(\mathbf{X}') = \beta \in [0, 1]$.

Note that, compared to the proof of Proposition 1, here we combined the last two items, resulting in only three items. In the first two cases, by the definition of terminal state, $\mathbf{X}'$ is also a terminal state with zero value. Also note that in the case of reachability, the reward integrates to one over an interval where $B$ occurs and is zero elsewhere. Similarly to the previous proposition, let $\mathcal{C}_1$ to $\mathcal{C}_3$ represent the sets of all possible states $\mathbf{X}'$ corresponding to each of the three cases above, and these

sets are mutually disjoint. Here, we show that if $V_\Lambda^*(\mathbf{x}) = 0$, then *only* either (2) can happen, or else (3) can happen, in which case $\beta$ has to be zero. We write

$$
\begin{aligned}
0 = V_\Lambda^*(\mathbf{x}) &= \mathbb{E}\Big[ \int_0^\Delta R_\Lambda(\mathbf{X}(t, \omega))\mathrm{d}t + V_\Lambda^*(\mathbf{X}'(\omega)) \mid \mathbf{X}(0, \cdot) = \mathbf{x} \Big] \\
&= \mathcal{P}[\mathbf{X}' \in \mathcal{C}_1]\big(1 + 0\big) + \mathcal{P}[\mathbf{X}' \in \mathcal{C}_2]\big(0 + 0\big) + \mathcal{P}[\mathbf{X}' \in \mathcal{C}_3]\big(0 + \beta\big) \\
&= \mathcal{P}[\mathbf{X}' \in \mathcal{C}_1] + \beta\mathcal{P}[\mathbf{X}' \in \mathcal{C}_3] \\
&= \Big(1 - \mathcal{P}[\mathbf{X}' \notin \mathcal{C}_1]\Big) + \beta\mathcal{P}[\mathbf{X}' \in \mathcal{C}_3] \\
&= 1 - \mathcal{P}[\mathbf{X}' \in \mathcal{C}_2 \cup \mathcal{C}_3] + \beta\mathcal{P}[\mathbf{X}' \in \mathcal{C}_3] \\
&= 1 - \mathcal{P}[\mathbf{X}' \in \mathcal{C}_2] + \big(-1 + \beta\big)\mathcal{P}[\mathbf{X}' \in \mathcal{C}_3]
\end{aligned}
$$

which deduces

$$
\mathcal{P}[\mathbf{X}' \in \mathcal{C}_2] + \big(1 - \beta\big)\mathcal{P}[\mathbf{X}' \in \mathcal{C}_3] = 1 \tag{13}
$$

Hence, the following cases are possible (note: $\mathcal{P}[\mathbf{X}' \in \mathcal{C}_k] = 1$ implies $\mathcal{P}[\mathbf{X}' \in \mathcal{C}_{k'}] = 0$, $k' \neq k$):

(i) $\mathcal{P}[\mathbf{X}' \in \mathcal{C}_2] = 1$;

(ii) $\mathcal{P}[\mathbf{X}' \in \mathcal{C}_3] = 1$ with $\beta = 0$ (otherwise the equality cannot hold);

(iii) both $\mathcal{P}[\mathbf{X}' \in \mathcal{C}_2] \neq 1$ and $\mathcal{P}[\mathbf{X}' \in \mathcal{C}_3] \neq 1$, with $\beta \neq 1$ (otherwise the equality cannot hold).

Note that if $\beta \neq 0$ then $\mathcal{P}[\mathbf{X}' \in \mathcal{C}_2]$ cannot be zero because $1 - \beta < 1$ and equation 13 would be violated. That is, in the case of $\mathcal{P}[\mathbf{X}' \in \mathcal{C}_3] = 1$ (hence, $\mathcal{P}[\mathbf{X}' \in \mathcal{C}_2] = 0$), $\beta$ has to be zero. On the other hand, if $\beta = 1$, then $\mathcal{P}[\mathbf{X}' \in \mathcal{C}_2]$ must be one; hence, in (iii), $\beta = 1$ must be excluded.

Let both $\mathcal{P}[\mathbf{X}' \in \mathcal{C}_2] \neq 1$ and $\mathcal{P}[\mathbf{X}' \in \mathcal{C}_3] \neq 1$. Using the fact that $\mathcal{P}[\mathbf{X}' \in \mathcal{C}_2] + \mathcal{P}[\mathbf{X}' \in \mathcal{C}_3] \leq 1$ and substituting from equation 13, it yields

$$
\mathcal{P}[\mathbf{X}' \in \mathcal{C}_2] + \frac{1 - \mathcal{P}[\mathbf{X}' \in \mathcal{C}_2]}{1 - \beta} \leq 1
$$

Re-arranging the terms and having note that $\beta \neq 1$, it follows that $1/(1 - \beta) \leq 1$ or $1 - \beta \geq 1$, which deduces $\beta = 0$. Substituting $\beta = 0$ in equation 13, it follows that

$$
\mathcal{P}[\mathbf{X}' \in \mathcal{C}_2] + \mathcal{P}[\mathbf{X}' \in \mathcal{C}_3] = 1
$$

which implies $\mathcal{P}[\mathbf{X}' \in \mathcal{C}_1] = 0$. Thus, occurrence of a terminal state that admits $B$ is improbable. Furthermore, in the case that both $\mathcal{P}[\mathbf{X}' \in \mathcal{C}_2] \neq 1$ and $\mathcal{P}[\mathbf{X}' \in \mathcal{C}_3] \neq 1$, $\beta$ must be zero. That is, the next state $\mathbf{X}'$ is (with probability one) either a terminal state *not* admitting occurrence of $B$, or else a non-terminal state with $V_\Lambda(\mathbf{X}') = 0$. Continuing with this line of argument and knowing that by assumption the time horizon is finite, we conclude that $V_\Lambda$ remains at zero until reaching a terminal state, which does not admit occurrence of $B$ (hence, $B$ never happens). Using value lemma, it then follows that $\Lambda_B$ also remains at zero and $B$ will never occur.

Conversely, if $B$ never happens, then all possible trajectories will incur zero return; thus, the expected return is zero, i.e., $V_\Lambda = 0$, which deduces $\Lambda_B = 0$.

$\square$

**Proposition 3.** *Let action $\boldsymbol{u}$ be selected according to some policy $\pi(\boldsymbol{x})$ over a time interval $[t_1, t_2]$. The resultant expected changes in grit and reachability of some future event $B$ are bounded as follows:*

1. $\min_\pi \mathbb{E}[\Delta\Gamma_B] \leq 0$

2. $\mathbb{E}[\Delta\Lambda_B] \leq 0$ *for all $\pi$*

*with the equality in both statements holds for deterministic environments.*

*Proof.* As the argument goes for any arbitrary point before event $B$'s occurrence, the reward of both MDP's are zero by definition. Also, remark that there is no discounting. Hence, the value functions in Lemma 1, $V_\Gamma^*$ and $V_\Lambda^*$, admit HJB equation of the following form:

$$\max_{\mathbf{u}} \left[ \sum_{j=1}^{n} \mu_j(\mathbf{x}, \mathbf{u}) \cdot \frac{\partial V_\Gamma^*}{\partial x_j}(\mathbf{x}) + \frac{1}{2} \sum_{i=1}^{n} \sum_{j=1}^{n} \boldsymbol{\sigma}_i(\mathbf{x}, \mathbf{u}) \boldsymbol{\sigma}_j^T(\mathbf{x}, \mathbf{u}) \cdot \frac{\partial^2 V_\Gamma^*}{\partial x_i \partial x_j}(\mathbf{x}) \right] = 0 \qquad (14)$$

where $\mathbf{x}$ is any state that does not admit the occurrence of $B$. From the value lemma we have $\Gamma_B(\mathbf{x}) = -V_\Gamma^*(\mathbf{x})$. Let $\pi$ denote any arbitrary stationary policy to select $\mathbf{u}$ (not necessarily fixed) from time $t_1$ to $t_2$. We have:

$$
\begin{aligned}
\min_{\pi} \mathbb{E}\left[\Delta\Gamma_B\right] &= \min_{\pi} \mathbb{E}\left\{ \int_{t_1}^{t_2} \left( \sum_{j=1}^{n} \mu_j(\mathbf{x}, \mathbf{u}) \cdot \frac{\partial \Gamma_B}{\partial x_j}(\mathbf{x}) + \frac{1}{2} \sum_{i=1}^{n} \sum_{j=1}^{n} \boldsymbol{\sigma}_i(\mathbf{x}, \mathbf{u}) \boldsymbol{\sigma}_j^T(\mathbf{x}, \mathbf{u}) \cdot \frac{\partial^2 \Gamma_B}{\partial x_i \partial x_j}(\mathbf{x}) \right) \mathrm{d}t \right\} \\
&= \max_{\pi} \mathbb{E}\left\{ \int_{t_1}^{t_2} \left( \sum_{j=1}^{n} \mu_j(\mathbf{x}, \mathbf{u}) \cdot \frac{\partial V_\Gamma^*}{\partial x_j}(\mathbf{x}) + \frac{1}{2} \sum_{i=1}^{n} \sum_{j=1}^{n} \boldsymbol{\sigma}_i(\mathbf{x}, \mathbf{u}) \boldsymbol{\sigma}_j^T(\mathbf{x}, \mathbf{u}) \cdot \frac{\partial^2 V_\Gamma^*}{\partial x_i \partial x_j}(\mathbf{x}) \right) \mathrm{d}t \right\} \\
&\leq \mathbb{E}\left\{ \int_{t_1}^{t_2} \max_{\mathbf{u}} \left( \sum_{j=1}^{n} \mu_j(\mathbf{x}, \mathbf{u}) \cdot \frac{\partial V_\Gamma^*}{\partial x_j}(\mathbf{x}) + \frac{1}{2} \sum_{i=1}^{n} \sum_{j=1}^{n} \boldsymbol{\sigma}_i(\mathbf{x}, \mathbf{u}) \boldsymbol{\sigma}_j^T(\mathbf{x}, \mathbf{u}) \cdot \frac{\partial^2 V_\Gamma^*}{\partial x_i \partial x_j}(\mathbf{x}) \right) \mathrm{d}t \right\} \\
&= 0 \qquad\qquad\qquad\qquad\qquad\qquad\qquad\qquad\qquad\qquad\qquad\qquad\qquad\qquad\qquad\qquad\qquad (15)
\end{aligned}
$$

The first line follows from decomposition lemma and the second line follows from value lemma. Remark that the negative sign in value lemma switches $\min$ to $\max$. Finally, the last line follows from equation 14. If the transitions are deterministic, then the expectation operators (as well as all the $\boldsymbol{\sigma}$ terms) will vanish. Hence, the inequality will also be replaced by an equal sign.

The proof for the second part follows a similar argument. We start with $\max_{\mathbf{u}} \mathbb{E}\left[\Delta\Lambda_B\right]$ and then apply the value lemma similar to the above (remark that there is no negative sign for reachability in the value lemma). This yields $\max_{\mathbf{u}} \mathbb{E}\left[\Delta\Lambda_B\right] \leq 0$, which induces $\mathbb{E}\left[\Delta\Lambda_B\right] \leq 0$. Hence, for reachability, the stated bound holds regardless of the choice of $\mathbf{u}$.

$\square$

**Proposition 4** (Causation). *Let $\mathcal{D}_A$ be the set of $A$'s ruling variables. $A$ is a cause of $B$ if and only if*

1. *$A$ happens before $B$;*

2. *$\mathbb{E}\{\Delta_A(\Gamma_B)\} > 0$;*

3. *$\sum_{j \in \mathcal{D}_A} \varphi_A(j) > -\sum_{j \notin \mathcal{D}_A} \min\left(\varphi_A(j), 0\right)$*

*Proof.* This follows from a direct translation of Definition 1 into our formal concepts as well as using decomposition lemma to bring the individual contributions.

$\square$

PROOF OF PROPERTIES FROM SECTION 4

i. **Efficiency:** The collective contribution of all components during any time interval is equal to $\Delta\Gamma_B$ over that interval.

   **Proof.** This is a direct result from decomposition lemma.

ii. **Symmetry:** If two variables are symmetrical w.r.t. $X_b$ (i.e., having exactly the same impact on the dynamics of other variables, which ultimately reach $X_b$), then switching them does not impact $\partial_j \Gamma_B$. Furthermore, their contributions in $\Delta\Gamma_B$ will be exactly the same provided that their respective $\mu$ and $\sigma$ are the same during the given time interval.

   **Proof.** It is immediate from equation 8 and decomposition lemma.

iii. **Null event:** Contribution of $X_j$ in $\Delta\Gamma_B$ is zero *if and only if* at some stopping time through the propagation chain of equation 8, $\mathcal{D}_k$ is empty (meaning that there is no link between $X_j$ at $t$ and $X_b$ at $t'$). Such an event is called *null event* w.r.t. $B$.

**Proof.** If $\boldsymbol{\mu}$ is non-zero for ruling variables of $A$, then decomposition lemma asserts that $\partial_j\Gamma_B$ must be zero in order to render $j$-th contribution null. Assuming that event $B$ is happening or has happened, the only way for that to become zero is that at least one of the $\mathcal{D}_k$'s in equation 8 is empty.

iv. **Linearity:** Let $A_i$, $A_j$, and $A_{i,j}$ be three events with the ruling variables $X_i$, $X_j$ and $\{X_i, X_j\}$, respectively. Then, the contribution of $A_{i,j}$ in $\Delta\Gamma_B$ is sum of the contributions of $A_i$ and $A_j$.

**Proof.** This follows from decomposition lemma.

**Propositions 5 and 6** These propositions summarize the explanation before them in a formal format. Note also that $A$ should first be a cause for $B$, then other conditions should be checked.

## C  COMPARISON TO THE HP DEFINITIONS

In this section, we first provide an overview of how our framework varies from the HP framework broadly.

We then discuss and understand the HP definition of actual causation (Halpern and Pearl, 2001; 2005; Halpern, 2015; 2016), then provide a mapping to move between the HP framework and our framework and then compare and contrast our definition of causation and the HP definition of actual causation.

### C.1  OVERVIEW

Broadly our framework differs from the HP framework in the following ways:

- Time is an explicit factor in our formulation. It establishes the direction of causation between events and answers questions of causation in more practical and complicated scenarios. Since we define events as changes in variables over a homogeneous interval of time, it clears much of the confusion around the time of happening of an event and provides the flexibility to study multiple events that share the same ruling variables and admit the same changes but occur at different points in time.

- Since we do not use interventions/manipulations to understand causation, we can make our conclusions from raw observational data without having to conduct interventions or worry about the kind of interventions, especially in complex systems.

- Instead of considering actions under a fixed policy, restricting the chain of events between $A$ and $B$, we examine if event $A$ causes $B$ under the most pessimistic version of such chains of events. Remark that adhering to a certain chain of actions can be readily considered in our framework. This respects the dynamics of the system between the events of interest and helps address more realistic scenarios.

- Further, we argue that an event can contribute partially to the happening of the effect without being necessary or sufficient. Our framework argues that, while a cause **can** be a sufficient and/or necessary cause, it does not **have** to be a sufficient/necessary cause to qualify as a possible cause of event $B$.

### C.2  STRUCTURAL EQUATION MODELLING

Before we look into the HP definitions we assume that the reader is familiar with the concept of structural equation modeling. To do so, we recommend the reader refer to (Beckers, 2021) section 2, so they are equipped with the basic tools needed to understand the HP definition. For our discussion, we will borrow heavily from this reference for notions and discussions necessary to define and understand structural causal modeling.

Much of the discussion and notation is taken from Halpern's actual causality (Halpern, 2016), as presented in Beckers (2021) with little change.

**Definition 2.** *(Beckers, 2021) A signature $\mathcal{P}$ is a tuple $(\mathcal{U}, \mathcal{V}, \mathcal{R})$, where $\mathcal{U}$ is a set of exogenous variables, $\mathcal{V}$ is a set of endogenous variables and $\mathcal{R}$ a function that associates with every variable $Y \in \mathcal{U} \cup \mathcal{V}$ a nonempty set $\mathcal{R}(Y)$ of possible values for $Y$ (i.e., the set of values over which $Y$ ranges). If $\vec{X} = (X_1, \ldots, X_n)$, $\mathcal{R}(\vec{X})$ denotes the cross product $\mathcal{R}(X_1) \times \cdots \times \mathcal{R}(X_n)$.*

It's important to recognize that exogenous variables are factors whose causes lie beyond the direct influence of the model, encompassing background conditions and noise. Conversely, the values of endogenous variables are causally influenced by other variables within the model, whether they are endogenous or exogenous. We refer to a collection $\vec{u} \in \mathcal{R}(\mathcal{U})$ of exogenous variable values as the contextual setup of the problem.

**Definition 3.** *(Beckers, 2021) A causal model $M$ is a pair $(\mathcal{P}, \mathcal{F})$, where $\mathcal{P}$ is a signature and $\mathcal{F}$ defines a function that associates with each endogenous variable $X$ a structural equation $F_X$ giving the value of $X$ in terms of the values of other endogenous and exogenous variables. Formally, the equation $F_X$ maps $\mathcal{R}(\mathcal{U} \cup \mathcal{V} - \{X\})$ to $\mathcal{R}(X)$, so $F_X$ determines the value of $X$, given the values of all the other variables in $\mathcal{U} \cup \mathcal{V}$.*

## C.3 HP Definitions

We now look at HP definitions of causation as presented in (Halpern, 2015). Halpern and Pearl (Halpern and Pearl, 2001; 2005) develop two of the initial HP definitions, whereas the third one is proposed solely by Halpern (Halpern, 2015). This will be the one we will discuss in detail since it builds on the limitations of the earlier proposed definitions. The relations between them are extensively discussed by Halpern Halpern (2016). We encourage the readers to read through this work to get a better insight into it.

We borrow on the notions used by (Beckers, 2021) in section 3 to discuss the HP definitions. Throughout our discussion, settings of variables $V$ with

- $*$ i.e., $\vec{v}^*$ indicate that $(M, \vec{u}) \models \left( \vec{V} = \vec{v}^* \right)$.
- $'$ i.e., $\vec{v}'$ indicate that $(M, \vec{u}) \models \left( \vec{V} \neq \vec{v}' \right)$.
- No subscripts can refer to any setting.

Given the notations, the modified HP definition Halpern (2015) is given as follows:

**Definition 4** (Modified HP (Halpern, 2015)). *Let event $\varphi$ be $\vec{Y} = \vec{y}$. $\vec{X} = \vec{x}$ is an actual cause of $\varphi$ in $(M, \vec{u})$ if the following conditions hold:*

AC1. $(M, \vec{u}) \models (\vec{X} = \vec{x}) \wedge \varphi$

AC2(a). *There is a partition of $\mathcal{V}$ into two sets $\vec{Z}$ and $\vec{W}$ with $\vec{X} \subseteq \vec{Z}$ and a setting $\vec{x}'$ and $\vec{w}$ of the variables in $\vec{X}$ and $\vec{W}$, respectively, such that $(M, \vec{u}) \models \left[ \vec{X} \leftarrow \vec{x}', \vec{W} \leftarrow \vec{w}^* \right] \neg \varphi$*

AC2(b). *For all subsets $\vec{Q}$ of $\vec{W}$ and subsets $\vec{O}$ of $\vec{Z} - \vec{X}$, we have $(M, \vec{u}) \models [\vec{X} \leftarrow \vec{x}, \vec{Q} \leftarrow \vec{q}, \vec{O} \leftarrow \vec{o}^*] \varphi$*

AC3. *$\vec{X}$ is minimal; there is no strict subset $\vec{X}''$ of $\vec{X}$ such that $\vec{X}'' = \vec{x}''$ satisfies AC2, where $\vec{x}''$ is the restriction of $\vec{x}$ to the variables in $\vec{X}''$*

We designate $\vec{W} = \vec{w}$ as a witness indicating that when $\vec{X} = \vec{x}$, it causes the occurrence of $\varphi$. The variables within $\vec{Z}$ are conceptualized as constituting the "causal path" from $\vec{X}$ to $\varphi$. In essence, altering the value of a variable in $\vec{X}$ leads to changes in the value(s) of certain variable(s) in $\vec{Z}$, subsequently influencing the values of other variable(s) within $\vec{Z}$, ultimately resulting in a change in the truth value of $\varphi$.

AC1 stipulates the basic criterion that both the potential cause and effect must be events that occurred. AC3 is similarly straightforward, emphasizing the exclusion of redundant elements from the causal factors. However, the crux of the definition lies in AC2. Halpern categorizes conditions AC2(a) and AC2(b) as the "necessity condition" and the "sufficiency condition," respectively (Halpern, 2015).

AC2(a) asserts that the effect demonstrates counterfactual dependence on the cause, holding the witness fixed at their actual values, i.e, $(M, \vec{u}) \models [\vec{X} \leftarrow \vec{x}, \vec{W} \leftarrow \vec{w}^*]\varphi$, and $(M, \vec{u}) \models [\vec{X} \leftarrow \vec{x}', \vec{W} \leftarrow \vec{w}^*]\neg\varphi$. Consequently, AC2(a) can be interpreted as articulating a contrastive necessity condition: there exist contrasting values $\vec{x}'$ such that altering $\vec{X}$ to $\vec{x}'$ results in non-fulfillment of $\varphi$.

The sufficiency condition, AC2(b), essentially stipulates that when the variables in $\vec{X}$ and any chosen subset $\vec{O}$ of other variables along the causal path (besides those in $\vec{X}$) retain their actual context values, then $\varphi$ holds even if only a subset $\vec{Q}$ of the variables in $\vec{W}$ are set to their values in $\vec{w}$ (the setting for $\vec{W}$ utilized in AC2(a)). Notably, by fixing $\vec{W}$ to $\vec{w}$, alterations in the values of variables within $\vec{Z}$ may occur. AC2(b) asserts that these changes do not impact the truth of $\varphi$; it remains valid. Consequently, in light of condition AC2(a), this implies that the variables in $\vec{W} - \vec{Q}$, denoted as $\vec{W}'$, essentially operate as they do in reality; their values are determined by the structural equations, denoted as $\vec{w}'^*$.

### C.4 Mapping from HP Formulation to our formulation

Mapping the problem from a static space to a dynamic space can be quite tricky because of the missing time component in the static setting. In fact, static formulations lacking time makes them particularly difficult to map into dynamical problems.

Recapping from Def. 2 and Def. 3, in HP framework, a causal model is formally defined as, $\mathcal{M} = (\mathcal{P}, \mathcal{F})$. $\mathcal{P}$ is a signature of tuple $(\mathcal{U}, \mathcal{V}, \mathcal{R})$, where $\mathcal{U}$ is a set of exogenous variables, $\mathcal{V}$ is a set of endogenous variables. $\mathcal{F}$ defines a function that associates with each endogenous variable $X$ a structural equation $F_X$ giving the value of $X$ in terms of the values of other endogenous and exogenous variables. Note that there are no functions associated with exogenous variables; their values are determined outside the model. We call a setting $\vec{u} \in \mathcal{R}(\mathcal{U})$ of values of exogenous variables a context. Because we are discussing causation in dynamical systems, we will restrict our discussion to strongly recursive (or strongly acyclic) causal models - where given context $\vec{u}$ the values of all remaining variables can be determined.

In our process-based formulation, the model under discussion is a Markov Decision Process (MDP), formally defined as a tuple $M = (\mathcal{S}, \mathcal{A}, R, \mathcal{P}_0)$. $\mathcal{S}$ and $\mathcal{A}$ are sets of possible states and actions (intervenable inputs), $R : \mathcal{S} \mapsto \mathbb{R}$ is a scalar reward function, and $\mathcal{P}_0$ is the distribution of initial states. We consider the state ($S(t) \in \mathcal{S}$), to be an $n$-dimensional vector space and is either fully observable or reconstructable from observations. At any given time, each state component ($S_j(t)$) is a random variable and the state vector's evolution across time forms a (stochastic) process. The evolution of state is, therefore, a function of both the intrinsic dynamics and the selected (extrinsic) actions across time. Therefore, the state component $S_j(t)$ can be seen as a function of other state and action components from prior times. Further, we say that $S(t)$ admits one or more known components $s_j$ at time $t$ iff $S_j(t) = s_j$.

As mentioned earlier, mapping from a static space to a dynamic space can be quite tricky because of the missing time component in the static setting. However, given our discussion so far, we can forge some associations.

The endogenous variables ($\mathcal{V}$) of the causal model $\mathcal{M}$ in HP setting can be analogous to the state variables of the MDP model $M$, but only at a certain time instance, $S(t)$ (n-dimensional vector of random variables). It's important to note that, since each time instance of any state variable is an endogenous variable, the causal graph in the static sense can grow immensely even for a small MDP - with only a few state variables and time steps. Because we are discussing strongly recursive causal models, the exogenous variables $\mathcal{U}$ in $\mathcal{M}$ can be thought of as providing the initial conditions, $\mathcal{P}_0$ in the MDP model $M$.

In our framework, we formally define an event as a change of one or more state or action components during a homogeneous time interval. The components involved in an event $A$ are called ruling variables of event $A$, or $\mathcal{D}_A$. State admits event $A$ between time $[t, t']$ if, $\mathcal{D}_A$ admits a certain change of values between $[t, t']$, say $S_{\mathcal{D}_A}(t') = \vec{x}$ and $S_{\mathcal{D}_A}(t)$ admits values other than $\vec{x}$. In HP setting (Halpern, 2016), given a causal model $\mathcal{M}$, a primitive event is a formula of the form $\vec{X} = \vec{x}$, for $X \in \mathcal{V}$ and $x \in \mathcal{R}(X)$. Since the HP definitions do not explicitly consider time in the definition of events, for simplicity let's assume that variables $\vec{X}$ admit values $\vec{x}$ only once during the observation period.

Therefore, $\vec{X} = \vec{x}$ implies $\mathcal{D}_A$, admits a certain change of values between $[t, t']$, say $S_{\mathcal{D}_A}(t') = \vec{x}$ (implying that in this case, for simplicity, we consider that $S_{\mathcal{D}_A}(t)$ admits some value of no consequential interest here). Note that we define every event w.r.t to both ruling variables and an associated time interval. This clears much of the confusion around the time of happening of an event and provides the flexibility to study multiple events that share the same ruling variables and admit the same changes but occur at different points in time.

As mentioned earlier, considering arbitrary policies for action selection, one may devise different chains of events after the cause event $A$. Following each such policy incurs a different probability of event $B$'s occurrence. In our setting, we examine if event $A$ causes $B$ under the most pessimistic version of such chains of events. In the HP setting, we have already seen that exogenous variables $\mathcal{U}$ provide the context under which the events occur. Therefore, we can also assume that exogenous variables in $\mathcal{U}$ along with providing initial condition $\mathcal{P}_0$ also define a fixed policy. So given a context,

| Our Formulation | HP Formulation |
|:---:|:---:|
| $S_{\mathcal{D}_A}(t_A)$ | $\vec{X}$ |
| $S_{\mathcal{D}_B}(t_B)$ | $\vec{Y}$ |
| $S_{\mathcal{D}_W}(t_W)$ | $\vec{W}$ |
| $S_{\mathcal{D}_Z}(t_Z)$ | $\vec{Z}$ |
| $S_{\mathcal{D}_{Z'}}(t_{Z'})$ | $\vec{Z}'$ |

Table 1: Mapping between variables in our framework and HP framework

$(\vec{u} \in \mathcal{R}(\mathcal{U}))$ the values of $\mathcal{A}$ are sampled based on a chosen policy. This is another distinction we hold to HP formulation, instead of considering actions under a fixed policy, restricting the chain of events between $A$ and $B$, we examine if event $A$ causes $B$ under the most pessimistic version of such chains of events. Remark that adhering to a certain chain of actions can be readily considered in our framework. In that case, the actions after event $A$ will simply become part of the dynamics and no longer exogenous variables.

In all HP definitions, the endogenous variables are divided into two disjoint subsets $\vec{Z}$ and $\vec{W}$. In these settings, the endogenous variables $\vec{X}$ defining a event ($\vec{X} = \vec{x}$) are a subset of endogenous variables $\vec{Z}$ (i.e., $\vec{X} \subset \vec{Z}$). Let's call the remaining endogenous variables of $\vec{Z}$, as $\vec{Z}'$ ($\vec{Z}$ - $\vec{X}$). Further, $\vec{Z}'$ is a set of endogenous variables that form a causal path between $\vec{X} = \vec{x}$ and the effect event, $\vec{Y} = \vec{y}$. Therefore, the complete set of endogenous variables $\mathcal{V}$ can be broken into 3 disjoint subsets - $\mathcal{X}, \mathcal{W}$ and $\mathcal{Z}'$.

As discussed before, endogenous variables can be mapped to the state variables of the MDP model $M$, but only at a certain time instance. For the purposes of this discussion, let's consider $\vec{X} = \vec{x}$ as event $A$ and $\varphi$ or $\vec{Y} = \vec{y}$ as event $B$. Therefore in our setting, $\vec{X}$ can be mapped to state components of ruling variables of the event of interest $A$ at time $t_A$, i.e., $S_{\mathcal{D}_A}(t_A)$. Similarly, $\vec{Z}'$ to $S_{\mathcal{D}_{Z'}}(t_{Z'})$. Since $\vec{Z}'$ are endogenous variables that form a causal path between event $\vec{X} = \vec{x}$ and the effect event, $\vec{Y} = \vec{y}$, we can assume that time of occurrence of these events $t_{Z'} \geq t_X$. Further, $\vec{W}$ can be denoted by state variables $S_{\mathcal{D}_W}(t_W)$. Since these variables do not form a causal path between cause and effect events, we can assume without harm that $t_W \leq t_X$. Even if some events defined by $\vec{W}'$, where $\vec{W}' \subset \vec{W}$, take place at $t_{W'} > t_X$, since they are not a part of the causal path between cause and effect events, they can safely be ignored for the purposes of our discussion. Similar conclusions can be drawn for the event $\vec{Y} = \vec{y}$, i.e, it is mapped to $S_{\mathcal{D}_B}(t_B)$. Since the effect event can only happen after the cause event $t_B > t_A$.

With these associations between the HP formulation and our formulation, we can move ahead to analyze the conditions of causation.

## C.5 COMPARE AND CONTRAST WITH HP DEFINITION

Given the current understanding of the HP definitions, let us now compare and contrast the modified HP definition of actual causation (Def. 4) with our definition of causation (Def.1). For the purposes of this discussion, let's consider $\vec{X} = \vec{x}$ as event $A$ and $\varphi$ or $\vec{Y} = \vec{y}$ as event $B$. Also, since the HP definitions do not explicitly consider time in the definition of events, for simplicity let's assume that variables $\vec{X}$ admit values $\vec{x}$ only once during the observation period. The same applies to $\vec{Y} = \vec{y}$. From discussed mapping in C.4, $\vec{X} = \vec{x}$ implies $S_{\mathcal{D}_A}$, admits a certain change of values between $[t_A, t'_A]$, say $S_{\mathcal{D}_A}(t'_A) = \vec{x}$ (implying that in this case, for simplicity, we consider that $S_{\mathcal{D}_A}(t_A)$ admits some value of no consequential interest here). Similar logic can be applied to event $B$.

**Our Definition:** In a stochastic process, we define event $A$ to be a cause of event $B$ if and only if:

C1. Time-wise, conclusion of $A$ happens at or before beginning of $B$;

C2. Expected grit of $B$ strictly increases from before to after $A$. Moreover, until $B$'s occurrence, it never becomes the same or smaller than its value at $A$'s beginning;

C3. The contribution of $A$'s ruling variables in the growth of $B$'s expected grit is strictly positive and is strictly larger in magnitude than that of non-ruling variables with negative impact.

**Condition AC1** AC1 represents the trivial requirement that the candidate cause and effect are among the events that took place. Our condition C1. implicity covers HP condition AC1 (Def. 4), adding additional clarity of direction of causation with a time arrow between the events.

**Condition AC2(a)** Condition AC2(a) states that for $\vec{X} = \vec{x}$ to be a cause, in causal model $\mathcal{M}$ with context $\vec{u}$ and witness $\vec{W}(u) = \vec{w}^*$, $\vec{Y} \neq \vec{y}$, if we set $\vec{X} = \vec{x}'$ ($\vec{x}' \in \mathcal{R}(\vec{\mathcal{X}})$ is value $\vec{X}$ does not take under the context $\vec{u}$). Note that $\vec{w}^*$ is the value $\vec{W}$ takes under $\vec{u}$ in causal model $\mathcal{M}$. This implies that there exist contrast values $\vec{x}'$ such that if $\vec{X}$ is set to $\vec{x}'$, $\varphi$ no longer holds. We further note that setting $\vec{X} = \vec{x}'$ also implies possible changes in values of $\vec{Z}$ that form a causal path between $\vec{X}$ and $\varphi$, ($\vec{Y} = \vec{y}$).

*Mapping to our framework*: Condition AC2(a) implies that event $A$ can be a cause of event $B$ if, for any other value of the ruling variables( $\vec{X} = \vec{x}'$) i.e, $S_{\mathcal{D}_A}(t) = \vec{x}'$ leads to non-occurance of event $B$, implying non-occurrence of event $A$. All this while holding events $S_{\mathcal{D}_W}(t_W)$ at the same values as they were when event $B$ occurs, while $S_{\mathcal{D}_{Z'}}(t_{Z'})$ can change in the subsequent time steps.

*Shortcomings*: This condition suffers from some major problems: 1) In a continuous setting, $\vec{x}'$ can take infinitely many values even within the range of $\mathcal{R}(\vec{\mathcal{X}})$. Further setting $\vec{X} = \vec{x}'$ (value not achievable under context $\vec{u}$) while holding $\vec{W} = \vec{w}^*$ (value achieved under context $\vec{u}$) while symbolically meaningful, can be impossible to achieve in most if not all dynamical systems of practical importance (Cartwright, 2007). For example, in the T1 diabetes example discussed in the experiment section, if event A is a spike in blood glucose levels to 120, what would event NOT A be? blood glucose spike to 80? 90? 100? Further, it would be impossible to make interventions and set the blood glucose to the desired values to understand their effect. When interventions are frequent and can take continuous values, finding patients with similar statistics but different interventions - to act as counterfactuals, would be like finding a needle in a haystack. 2) More importantly, under this condition, it is implied that when a cause event does not happen, then the effect event does not happen as well. This enforces that, for an event to be a cause it should also be a necessary cause. This contradicts our understanding of causation where we argue that an event can contribute partially to the happening of the effect without being necessary or sufficient. Our framework argues that, while a cause **can** be a necessary cause, it does not **have** to be a necessary cause to qualify as a possible cause of event $B$.

*Comparison*: We argue that the intent of condition AC2(a) in Def.4 is to understand the effect of event $A$ ($\vec{X} = \vec{x}$ or $S_{\mathcal{D}_A}(t_A) = \vec{x}$) in isolation. This condition looks at the effect of the non-happening of event $A$ ($\vec{X} = \vec{x}'$ or $S_{\mathcal{D}_A}(t_A) = \vec{x}'$) on event $B$, while witness $\vec{W}$ are held under the values of context $\vec{u}$, i.e, when event $B$ ($\vec{Y} = \vec{y}$) occurs. Our conditions C2 and C3 address this issue without having to take an interventionist approach to causation. In our formulation, we can compute the contribution of each event (defined by a subset of the state/action components over a homogenous time interval) towards increasing the minimum probability of the event $B$ happening. This helps us talk about the individual contribution of each event of interest without the need to hold $S_{\mathcal{D}_W}(t_W)$ at the value achieved under context $\vec{u}$ while admitting a different value at $S_{\mathcal{D}_A}(t_A)$, which might be physically impossible to achieve.

**Condition AC2(b)** The sufficiency condition, as discussed in condition AC2(b) of Modified HP (Def. 4) roughly speaking mentions that if the variables in $\vec{X}$ and an arbitrary subset $\vec{O}$ of other variables on the causal path are held at their values in the actual context $\vec{u}$) (i.e, $\vec{X}(u) = \vec{x}$ and $\vec{O}(u) = \vec{o}^*$), then $\varphi$ holds even if any subset of $\vec{W}, \vec{Q}$ is set to $\vec{q}^*$. Therefore, this condition implies that once event $\vec{X} = \vec{x}$ happens, then no matter what values the events on the causal path between event $\vec{X} = \vec{x}$ and event $\varphi$, ($\vec{Y} = \vec{y}$) take, event $\vec{Y} = \vec{y}$ still happens.

*Mapping to our framework*: Event $A$ can be a cause of event $B$ if, no matter the values the non-ruling variables (action or state variables), affected by event $A$ and present in the causal path between event $A$ and event $B$ take in the subsequent time steps, event $B$ will still happen.

*Shortcomings*: Similar to the previous case, this condition enforces that, for an event to be a cause it should also be a sufficient cause. This again contradicts our understanding of causation where we argue that an event can contribute partially to the happening of the effect without being necessary or sufficient.

*Comparisons* Our framework argues that, while a cause **can** be a sufficient cause, it does not **have** to be a sufficient cause to qualify as a possible cause of event $B$.

**Condition AC3**    AC3 is also fairly straightforward: we should not consider redundant elements to be parts of causes. Further, any possible redundancies in ruling variables can be examined and eliminated, because, their individual contributions to the growth of B's expected grit will be zero. Hence we can also satisfy condition AC3.

# D   EXPERIMENT 1: ATARI GAME OF PONG

Both our network architecture and the base pipeline have the same structure as the original DQN paper Mnih et al. (2015). In particular, there are 3 convolutional layers followed by 2 fully-connected linear layers. The first convolutional layer has $32\ 8 \times 8$ filters with stride 4, the second $64\ 4 \times 4$ filters with stride 2, and the third and final convolutional layer contains $64\ 3 \times 3$ filters with stride 1. Then, the first linear layer has 512 inner nodes, and the next linear layer maps these to the number of actions. All layers except the last one are followed by ReLU nonlinearities (the last layer is just a linear layer). The state includes 4 consecutive frames, each of which is a downsized of the actual Atari screen into $84 \times 84$ pixels and then switched into grayscale. Thus, the actual state is a tensor of size $4 \times 84 \times 84$.

**Important point:** The screens shown in the main paper's Fig. 1 illustrate the last frame of these four at each step.

For the optimizer, we used Pytorch's implementation of Adam optimizer with the Huber loss function, and the results are obtained after training over 200 epochs of 250,000 steps each (each action is repeated 4 times, hence each epoch involves one million Atari frames). All the hyper-parameters are chosen similarly to Mnih et al. (2015).

In the plots, $\nabla\Gamma(\mathbf{x})$ is colour-coded by red shades for $\nabla\Gamma(\mathbf{x}) > 0$, blue shades for $\nabla\Gamma(\mathbf{x}) < 0$ and white for zero, with darkest red for $\nabla\Gamma(\mathbf{x}) \geq +1$ and darkest blue for $\nabla\Gamma(\mathbf{x}) \leq -1$. Values with $|\nabla\Gamma(\mathbf{x})| < 0.1$ are set to zero to denoise. The plots for $g$ are similar but using $0.05$ rather than 1 to magnify the presentation. Additionally, $g$ plots only depict $g \geq 0$; since, by definition, only a positive change of grit induces a cause.

Full details can be found in the `./atari` folder of the code, available at `https://github.com/fatemi/dynamical-causality`.

# E   EXPERIMENT 2: TYPE-1 DIABETES

## E.1   SETUP AND DETAILS

We use an open-source implementation[2] of the FDA-approved Type-1 Diabetes Mellitus Simulator (T1DMS) Kovatchev et al. (2009) for modeling the dynamics of Type-1 diabetes. In version is the very first release of the simulator and assumes inter-subject variability to be the same (parameters are sampled from a distribution with same covariance matrix) and does not model intra-day variability of patient parameters which they do in future iterations Man et al. (2014); Visentin et al. (2018).

The simulator models an insilicopatient's blood glucose level (BG) and 12 other body dynamics with real-valued elements representing glucose and insulin values in different compartments of the body. The glucose dynamics are captured by - plasma glucose, tissue glucose, glucose in stomach 1 (GS1), glucose in stomach 2 and gut. The insulin dynamics are captured by - insulin on glucose production, insulin on glucose utilization, insulin action on liver, plasma insulin, liver insulin, sub-cutaneous insulin-1 (SI) and sub-cutaneous insulin-2. We control two actions: 1) Insulin intake, to regulate the amount of insulin 2) Meal intake, to regulate the amount of carbohydrates.

Meal intake increases the amount of glucose in the bloodstream, for T1D patients, when unregulated without external insulin, this can lead to hyperglycemia - generally characterized by blood glucose levels shooting over 180 mg/dL. Tight blood glucose control with insulin injections can help, but intensive control of blood glucose with insulin injections can increase the risk of hypoglycemia - generally characterized by blood glucose levels less than 70 mg/dL.

For our experiment, we study the event of hypoglycemia (BG < 70 mg/dL).

**Insulin Dosing and Intake:**   Insulin dosing in T1D will vary based on the patient's age, weight, and residual pancreatic insulin activity. T1D patients will typically require a total daily insulin dose of 0.4 - 1.0 units/kg/day. For example, if a patient weighs 80 kg, the total daily dose = 80 kg X (0.5 units/kg/d) = 40 units per day. Typically this insulin is split into basal and bolus insulin. Usually,

---

[2]`https://github.com/jxx123/simglucose`

basal insulin is the insulin taken to keep the blood glucose in a steady state when there is no meal intake. It is generally regulated through an insulin pump. Bolus insulin on the other hand is taken through insulin injections to explicitly regulate the rise in glucose levels with meal intake. For the purpose of our simulation, we focus on bolus insulin intake only. We consider a constant intake of 0.027 units/min of basal insulin. For our experiment bolus insulin intake takes on values from $A_{ins} = \{0, 3, 7, 15\}$ units.

We note that since this version of the simulator does not model the intra-day variability of the patient parameters, the patient reaction to a given action is not dependent on the time of the day, i.e, 3 units of insulin intake at noon and 3 units of insulin intake at 6 pm should have the same effect on patient dynamics. To generate realistic daily insulin scenarios, we use a random scenario generator. Each scenario is associated with the time of intake, amount of intake, and probability of intake. To make the daily insulin intake more realistic, we add some stochasticity to the time of insulin intake, by modeling it with a truncated normal distribution, $TN(\mu, \sigma, lb, ub)$ centered around time $\mu$ with $\sigma$ variance, $lb$ lower bound and $ub$ upper bound. choosing different probabilities of consumption helps generate realistic scenarios of excessive insulin intake leading to hypoglycemia, or excessive meal intake in the absence of insulin leading to hyperglycemia. We design six possible intakes of insulin roughly capturing insulin intakes at breakfast, snack1, lunch, snack2, dinner, and snack3.

| Time | Amount (in units) | Probability of Consumption | |
|---|---|---|---|
| $TN(3, 1, 1, 5)$ | 7 | $\{0.95, 1\}$ | |
| $TN(5.5, .5, 5, 6)$ | 3 | 0.3 | |
| $TN(8, 1, 6, 10)$ | 15 | 1 | (16) |
| $TN(11, .5, 10, 12)$ | 3 | 0.3 | |
| $TN(14, 1, 12, 16)$ | 15 | 1 | |
| $TN(17.5, 1, 16, 19)$ | 3 | 0.3 | |

**Meal Intake:** In our simulation, meal intake can take 4 different possible values, $A_{meal} = \{0, 5, 30, 60\}$ grams. Similar to insulin intake, to generate realistic daily meal consumption scenarios, we use a random scenario generator. Each scenario is associated with the time of intake, amount of intake, and probability of intake. To add some stochasticity to the time of meal intake, by modeling it with a truncated normal distribution, $TN(\mu, \sigma, lb, ub)$ centered around time $\mu$ with $\sigma$ variance, $lb$ lower bound and $ub$ upper bound. We design six possible intakes of meals roughly capturing breakfast, snack1, lunch, snack2, dinner, and snack3.

| Time | Amount (in grams) | Probability of Consumption | |
|---|---|---|---|
| $TN(3.5, 1, 1, 5)$ | 30 | $\{0.95, 0\}$ | |
| $TN(6, .5, 5, 6)$ | 5 | $\{0.3, 0.5\}$ | |
| $TN(8.5, 1, 6, 10)$ | 60 | $\{0.5, 0, 1\}$ | (17) |
| $TN(11.5, .5, 10, 12)$ | 5 | 0.3 | |
| $TN(14.5, 1, 12, 16)$ | 50 | $\{0.5, 1\}$ | |
| $TN(18, 1, 16, 19)$ | 5 | 0.95 | |

We map every combination of insulin and carbohydrate intake (each of which takes 4 distinct values) into 16 different possible actions. For the purpose of our experiment, we sample from a single patient trajectory (adult003) over 24 hours with a 1-minute sampling interval. We start from the same initial conditions because the system becomes chaotic for different perturbations in initial conditions.

For our study, we use Monte Carlo estimation to estimate $V_\Gamma(\mathbf{X})$ and therefore $\Gamma_B(\mathbf{X})$. Given the exploration policy of the off-line data we generate using the simulator is close to optimal, we can safely assume this gives us $V_\Gamma^*(\mathbf{X})$. We set $\gamma = 1$ with no positive rewards, i.e., $r = -1$ if $BG < 70$ and zero otherwise. We terminate the episode if we hit either hyperglycemia (BG>180), or hypoglycemia (BG<70) or we reach the end of 24 hours. We use Pytorch's *autograd* to compute the value function's gradient w.r.t. different body dynamics, based on which we could compute g-formula with $M = 50$ computational micro-steps to compute the integral. In our setting, we observe that intake of insulin causes an instantaneous spike in dynamics of subcutaneous insulin 1 (SI1), while intake of carbohydrates causes an instantaneous spike in glucose in stomach 1 (GS1). Since the previous action is considered a part of the current state to keep the system Markovian, the spike in subcutaneous insulin just acts as a proxy for action insulin, similarly for action meal and GS1.

We use a simple deep network with 3 fully connected layers with GELU (Gaussian error linear unit) activation. In particular, we have $13 \times 30$, $30 \times 30$, and $30 \times 1$ fully connected layers. The actual state is a tensor of $13 \times 1$. Since the signals have very different ranges, we normalize them (with population mean and standard deviation) before passing them through the network.

We use a learning rate of 0.00001 and minibatch size of 128. In each minibatch, we select 64 transitions by sampling from a prioritized experience replay (PER) butter Schaul et al. (2015). For the remaining 64 samples, we choose 32 samples uniformly from the train data and append it with 6 uniformly selected event B, hypoglycemia transitions ($r = -1$ and terminal state = True), 2 uniformly selected hyperglycemia transitions ($r = 0$ and terminal state = True), and remaining samples are sampled from non-zero action samples. All other chosen hyper-parameters can be found in the config.yaml file in the root directory of our code.

### E.2   SYSTEM DYNAMICS MODEL

We use the ODE's provided by Kovatchev et al. (2009) to generate the signals. We notice that some of these are different from the ones provided in the open-source implementation repository simglucose and make the changes accordingly in our implementation.

#### E.2.1   THE ORAL GLUCOSE SUBSYSTEM THAT CONTROLS RATE OF GLUCOSE APPEARANCE

Stomach compartment 1

$$
\begin{aligned}
x^1(t) = \dot{Q}_{sto1}(t) &= -k_{max} \cdot Q_{sto1}(t) + D \\
D &= \text{CHO - carbohydrate intake from meal consumption}
\end{aligned}
\tag{18}
$$

Stomach compartment 2

$$
x^2(t) = \dot{Q}_{sto2}(t) = -k_{gut}(Q_{sto}) \cdot Q_{sto2}(t) + k_{max} \cdot Q_{sto1}(t)
\tag{19}
$$

Gut

$$
x^3(t) = \dot{Q}_{gut}(t) = -k_{abs} \cdot Q_{gut}(t) + k_{gut}(Q_{sto})
\tag{20}
$$

$$
Q_{sto}(t) = Q_{sto1}(t) + Q_{sto2}(t)
\tag{21}
$$

Glucose rate of appearance

$$
Ra(t) = \frac{f \cdot k_{abs} \cdot Q_{gut}(t)}{BW}
\tag{22}
$$

#### E.2.2   GLUCOSE SUBSYSTEM - GLUCOSE KINETICS

Plasma Glucose

$$
x^4(t) = \dot{G}_p(t) = \max(EGP(t), 0) + Ra(t) - U_{ii}(t) - E(t) - k_1 \cdot G_p(t) + k_2 \cdot G_t(t)
\tag{23}
$$

Tissue Glucose

$$
x^5(t) = \dot{G}_t(t) = -U_{id}(t) + k_1 \cdot G_p(t) - k_2 \cdot G_t(t)
\tag{24}
$$

Subcutaneous Glucose

$$
\begin{aligned}
x^{13}(t) = \dot{G}_s(t) &= -\frac{1}{T_s} \cdot G_s(t) + \frac{1}{T_s} \cdot G_p(t) \\
\dot{G}_s(t) &= (G_s(t) \geq 0)(\dot{G}_s(t)) \\
G(t) &= G_p(t)/V_G
\end{aligned}
\tag{25}
$$

Endogenous glucose production
$$EGP(t) = k_{p1} - k_{p2} \cdot G_p(t) - k_{p3} \cdot X_L(t)$$

Insulin independent utilization
$$U_{ii}(t) = F_{cns} \tag{26}$$

Insulin dependent utilization
$$U_{id}(t) = \frac{(V_{m0} + V_{mx} \cdot X(t)) \cdot G_t(t)}{K_{m0} + G_t(t)}$$

### E.2.3 INSULIN KINETICS

Plasma Insulin
$$x^6(t) = \dot{I}_p(t) = -(m_2 + m_4) \cdot I_p(t) + m_1 \cdot I_l(t) + k_{a1} \cdot I_{sc1}(t) + k_{a2} \cdot I_{sc2}(t) \tag{27}$$

Insulin action on glucose utilization
$$x^7(t) = \dot{X}(t) = -p_{2U} \cdot X(t) + p_{2U} \left( I(t) - I_b \right) \tag{28}$$

Delayed insulin action in the liver
$$x^8(t) = \dot{X}_L(t) = -k_i \cdot (X_L(t) - I(t)) \tag{29}$$

Insulin action on glucose production
$$I(t) = \frac{I_p(t)}{V_I}$$
$$x^9(t) = \dot{\tilde{I}}(t) = -k_i \cdot (\tilde{I}(t) - I(t)) \tag{30}$$

Liver insulin
$$x^{10}(t) = \dot{I}_l(t) = -(m_1 + m_3) \cdot I_l(t) + m_2 \cdot I_p(t) \tag{31}$$

Subcutaneous insulin 1
$$x^{11}(t) = \dot{I}_{sc1}(t) = -(k_d + k_{a1}) \cdot I_{sc1}(t) + Insulin(t) \tag{32}$$

Subcutaneous insulin 2
$$x^{12}(t) = \dot{I}_{sc2}(t) = k_d \cdot I_{sc1}(t) - k_{a2} \cdot I_{sc2}(t) \tag{33}$$

