# OpenReview forum: "A Dynamical View of the Question of Why"
_ICLR.cc/2024/Conference — ICLR 2024 poster_

### Official Review · Reviewer_6KUF · 2023-10-26

**Soundness:** 2 fair
**Presentation:** 2 fair
**Contribution:** 2 fair
**Rating:** 5
**Confidence:** 4

**Summary:**

In this paper the authors propose a general theory for causation in dynamical systems, different from more mainstream causality models such as those of Spirtes and Pearl.

**Strengths:**

The review will be contained to this section owing to the flow in which it was conducted.

### Abstract

- I think you are being a bit unfair to the research community with your first sentence; there is plenty of work which considers causal reasoning in time, it is a basic tenet of causality (see Hume). Or I may have misunderstood the claim. Please explain or re-write.

### Introduction

- "If cause event A had not occurred, effect event B would not have occurred" this sentence does not make grammatical sense.
- "However, these approaches can have severe limitations, especially when applied to dynamical systems" - what are these limitations? Pearl has multiple works on dynamical plans, clearly then it is feasible to use his SCM for studying dynamics, hence please be more precise with your critique or shortcomings of his and Spirtes' approach.
- "Interventions are often infeasible in physical systems" - why is that? There are plenty of works to the contrary.
- "Moreover, these approaches ignore the dynamics as well as the possibility of other interventions between events" - but there's plenty of work in this area, see e.g. https://arxiv.org/pdf/1803.08784.pdf
- "Further, these frameworks assume knowledge of causal dependencies or structural information between various events in the system" but that is the core assumption of causal inference (notwithstanding causal discovery) that the causal diagram is known and usually that causal sufficiency holds at that. How can that be a critique of causal modelling in a paper about: causal modelling? Again, please elucidate on what you mean in this context.
- Very interesting third paragraph re: Salmon's work.
- I do not follow, are you suggesting that the example 'smoking causes lung cancer' is a poorly understood cause-effect relationship? Or are you suggesting that there are other causes which are not modelled by that same model?
- This seems like a contradiction wouldn't you agree: "real-world experiments conducted on simulated data".

### Basics and problem formulation

- "Any dynamical system can be described from the state viewpoint" this is categorically not true using Kalman's definition, for chaotic dynamics.
- Your first paragraph is essentially a re-statement of Pearl's SCM.
- What is a "filtered probability space" with emphasis on _filtered_?
- "More formally, under sufficient smoothness conditions" - and what are these conditions?
-  Why would limits (1) and (2) exist? Where do they come from? What is the reasoning? What's the deduction of their existence?
- Under process-based causality, I feel that several references are amiss. Yes those axioms (I and II) are mostly true (though (i) appears not to be within the space of quantum dynamics but let's not go there) but they would do well to be referenced. Hume argued the temporal priority principle a couple of hundred years ago, so this is nothing new and hence should be placed within the literature. Where does (iii) come from?
- You haven't specified what 'HP' stands for so 'HP settings' makes no sense.
- "central idea behind Lewis definition" who is Lewis here? What's the reference to?
- "causes, by themselves, increase the probability of their effects" that is nonsensical; are you arguing that if a cause exists then the effect _may_ happen?
- "as the role of intervention is to mechanically separate the impact of a variable from the collective impact" what does that mean?
- Typo: 'stochiastic process'
- What is a "sticky value"?

### Formal establishment of causation

- The grammar is getting noticeably worse as the paper is progressing. Consider proof-reading it a few times before next revision.
- So far I do not following the reasoning in this section, specifically though why would this be true "This could prove useful in the context of causal discovery."?
- Typo: "Equation equation 10 shows"
- "Wrongly identifying correlations as causal links is a core problem in formal reasoning" I would argue that it is a core problem in all of science.
- I suggest you pick anything but 'g-formula' as 'g-formula' is already a popular approach for estimating treatment or exposure effects from longitudinal data that are subject to time-varying confounding. Which is not what you are doing here and so because that name is already reserved it may be wise to select something more distinctive.

### Experiments

- "Modeling dynamical systems as SCMs is computationally and memory intensive to a prohibitive degree, especially in systems with numerous variables" this is not true and plenty of work do precisely this, here is a good review paper: https://arxiv.org/pdf/2001.06208.pdf
- I do not understand what we are looking at in figure 1. There is almost no explanation for what the axis metrics are, what the frames are, why they are chosen and what they are meant to represent. What is blue? What is red? How are we meant to understand the horizontal row of subplots for $g$? How is any of this explainable? Certainly you say that we do not need system equations and graphs for this method, but I am struggling to understand how this is at all more useful?
- Finally I do not agree with this statement at all "Consequently, existing methods are irrelevant for baseline comparisons" - if they are truly inadequate, as you claim, for dynamical settings then empirically show us why and show us how much better this theory is in comparison. Right now, unfortunately, we have to take you at face value that this statement is in fact true even though, in this review alone, multiple sources have been provided which point to the contrary. I do not think this is a fair evaluation of this work, let alone other models of causation.
- The captions for all the figures in this section are much much too short to describe the amount of detail that is happening not just in the whole figure but all the panels included therein. Regrettably the main text does a fairly poor job of providing the missing explanation as we ll.

**Weaknesses:**

See the main review in the above section.

**Questions:**

All my questions are in the Strengths section but I will just remark that I have rarely had so many questions after having reviewed a paper.

---

> ### Author Response · Authors · 2023-11-14
>
> We thank the reviewer for the feedback. Here, we address all the points.
>
> > I think you are being a bit unfair to the research community with your first sentence …
>
> We covered the literature on both static and dynamic settings in the Related Work section (we will update that section in light of new pointers). What we meant by the opening sentence is that causation in dynamical settings are far less attended to in the existing literature. If this still sounds unfair, we will update it to clarify.
>
> > "If cause event A had not occurred, effect event B would not have occurred" this sentence does not make grammatical sense.
>
> This sentence is both correct and accurate. Indeed, you can see this repeatedly in various philosophical texts when talking about counterfactual analysis. See for example Paul’s classic paper, 1st page right under “Lewis’s Analysis” (that’s why we have placed this statement inside quotation marks):
>
> L. A. Paul, ‘Problems with Late Preemption’, Analysis, vol. 58, no. 1, pp. 48–53, 1998.
>
> To help rectifying your mind, we italicise *cause event A*.
>
> The reviewer can further refer to this work to better understand this fundamental philosophy of causation: Menzies, Peter and Helen Beebee, "Counterfactual Theories of Causation", The Stanford Encyclopedia of Philosophy (Winter 2020 Edition), Edward N. Zalta (ed.), URL = <https://plato.stanford.edu/archives/win2020/entries/causation-counterfactual/>
>
> > what are these limitations? Pearl has multiple works …
>
> We would like to point out that the next few sentences of the paragraph highlight these with references. However, we will further discuss here some of the major differences along with the shortcomings of the interventionist/manipulationist framework:
>
> 1. Difficulty/impossibility of interventions in many dynamical settings: Unlike the interventions/manipulations approach to understand causation, we can make our conclusions from raw observational data without having to conduct interventions or worry about the kind of interventions, especially in complex systems. Further, interventions might not be possible in all physical systems, as we’ll discuss in detail in the answer to the next question.
>
> 2. Need to know causal graphs (a strong assumption) and scaling poorly with the number of variables. We discuss this in detail w.r.t the Atari example, in the response to another question of the reviewer.
>
> 3. Confusion about time of happening an event: our definition of an event clears all the confusion around the time of happening of an event and provides the flexibility to study multiple events that share the same ruling variables and admit the same changes but occur at different points in time.
>
> 4. Other interventions between events: Instead of considering actions under a fixed policy, i.e., restricting the chain of events between $A$ and $B$, we examine if event $A$ causes $B$ under the most pessimistic version of such chains of events. Remark that adhering to a certain chain of actions can also be readily considered in our framework.
>
> 5. These approaches ignore the dynamics as well as the possibility of other interventions between events (as explained later).
>
> In order to get specific differences between our framework and the Halpern and Pearl framework for causation [Joseph Y Halpern and Judea Pearl. Causes and explanations: A structural-model approach. part ii: Explanations] please refer to the last section of Appendix C.3.

---

> > ### Author Response · Authors · 2023-11-14
> > **Cont.**
> >
> > > intervention… why is that? There are plenty of works to the contrary
> >
> > While interventions make sense symbolically, atomic interventions as demanded by pearl’s framework might not be possible in all physical systems. For example, as Nancy Cartwright argues in “Hunting Causes and Using Them” (which we have cited in the paper), Cartwright provides a 4-equation model of a car carburetor and concludes:
> >
> > “The gas in the chamber is the result of the pumped gas and the gas exiting the emulsion tube. How much each contributes is fixed by other factors: for the pumped gas both the amount of airflow and a parameter $a$ , which is partly determined by the geometry of the chamber; and for the gas exiting the emulsion tube, by a parameter $a’$, which also depends on the geometry of the chamber. The point is this. In Pearl’s circuit-board, there is one distinct physical mechanism to underwrite each distinct causal connection. But that is incredibly wasteful of space and materials, which matters for the carburetor. One of the central tricks for an engineer in designing a carburetor is to ensure that one and the same physical design - for example, the design of the chamber - can underwrite or ensure a number of different causal connections that we need all at once. Just look back at my diagrammatic equations, where we can see a large number of laws all of which depend on the same physical features - the geometry of the carburetor. So no one of these laws can be changed on its own. To change any one requires a redesign of the carburetor, which will change the others in train. By design the different causal laws are harnessed together and cannot be changed singly. So modularity fails.”
> >
> > This is what we imply by “interventions might not be possible in all physical systems”, which further restricts them being used in practical real-world data.
> >
> > > "Moreover, these approaches ignore the dynamics as well as the possibility of other interventions between events" - but there's plenty of work in this area, see e.g. https://arxiv.org/pdf/1803.08784.pdf
> >
> > Popular interventionist/manupalist approaches to causation rely on the couterfactuals to make causal conclusions. For example under such frameworks, in order to understand if a treatment A1 (event A) is cause of patient death (event B) - it involves analyzing the counterfactual where -  if treatment A1 was not taken (event not A) by the patient, would it still cause their death (event not B), under the situation that the rest of the world is held at its previous state, i.e, when event A and event B have happened. In this understanding of causation, the possible dynamics between treatment A1 and the patient's death are ignored. In this example, it is possible that various other treatments, or interventions could be taken between treatment A1 and patient’s death, which might act to prevent the death of the patient or help accelerate it. Moreover, the metabolism of the patient's body may change due to other reasons and factors (quite common in the ICU conditions) which can accelerate or decelerate mortality. These frameworks fail to consider these dynamics.
> >
> > The work cited by the reviewer talks about atomic interventions in dynamic systems which are designed as structural dynamic causal models (SDCMs) that explicate the causal semantics of the system’s components, extending the framework of structural causal models (SCMs). Firstly, as discussed above, while it is symbolically meaningful, such interventions might not be possible in all physical systems. Further, understanding the effect of such interventions as mentioned in the referenced paper needs an understanding of structural relationships between all variables, which might be impossible in systems with a huge number of interacting variables. Moreover, the referenced paper looks at interventions in dynamic systems; it however does not consider any dynamics between the intervention and any possible effect event. We request the reviewer to please point out specifics in the paper if they believe we are missing something.

---

> > > ### Author Response · Authors · 2023-11-14
> > > **Cont.**
> > >
> > > > "Further, these frameworks assume knowledge of causal dependencies or structural information between various events in the system" but that is the core assumption of causal inference (notwithstanding causal discovery) that the causal diagram is known and usually that causal sufficiency holds at that. How can that be a critique of causal modelling in a paper about: causal modelling? Again, please elucidate on what you mean in this context.
> > >
> > >
> > > Not all works that try to understand and exploit causal relationships need access to causal graphs (and certainly not ours). For example, in popular work for causal representation learning, "Invariant risk minimization" by Arjovsky, Martin, et al., the aim is to learn causal representations by using data across environments without the need for causal graphs. As mentioned in this paper, while several causal representation learning works assume access to causal graphs - which can be a limiting assumption, and scales poorly with number of variables - this work aspires to learn causal relationships without the need for causal graphs. This work builds on the idea that causal relationships are invariant across environments in order  to learn causal representations.
> > >
> > > Similarly, we aim to learn causal relationships between events without needing causal graphs, by advocating for system level thinking. Our work proposes a dynamical systems approach to causation that builds on a philosophical framework based on the process theory of causation. A look at works such as “A dynamical systems approach to causation” by Peter Fazekas, et al., as cited in our related work section might help the reviewer better appreciate and understand the philosophical framework behind our proposed theory.
> > >
> > > > Very interesting third paragraph re: Salmon's work.
> > >
> > > Thank you. This paragraph emphasizes an alternative philosophy to understand the concept of causation apart from the popular interventionist/manipulationist approaches. It discusses frameworks that discuss a process based philosophy of causation that inspires our own framework.
> > >
> > > > I do not follow, are you suggesting that the example 'smoking causes lung cancer' is a poorly understood cause-effect relationship? Or are you suggesting that there are other causes which are not modelled by that same model?
> > >
> > >
> > > We are suggesting that the problem of lung cancer and the impact of smoking on it involves numerous variables (from systems biology) and an extremely complex dependency among such variables, both are completely unknown, let alone decomposing them into sets of individual interactions.
> > >
> > >
> > > > This seems like a contradiction wouldn't you agree: "real-world experiments conducted on simulated data"
> > >
> > > Thanks for the pointer; we corrected it to avoid confusion. By this, we implied that we work with simulated data that emulates real-world data (in the Diabetes example).
> > >
> > > > "Any dynamical system can be described from the state viewpoint" this is categorically not true using Kalman's definition, for chaotic dynamics.
> > >
> > > This statement is true and is what Kalman says in his paper (1960, cited in our paper). Being chaotic has nothing to do with the definition of state, as chaos is about perturbation of initial conditions causing unbounded change in the trajectory (chaos is a property of system's dynamics, not state).
> > >
> > > >  Your first paragraph is essentially a re-statement of Pearl's SCM.
> > >
> > > We respectfully do not understand your point. This paragraph sets the basic definitions and also talks about diffusion processes. Is this a re-statement of SCM?! We strongly believe it is not.
> > >
> > > > What is a "filtered probability space" with emphasis on filtered?
> > >
> > > It is a basic mathematical term (can be found in any textbook on stochastic calculus), meaning that the $\sigma$-algebra at any time is a superset of those of all the prior times.
> > >
> > > > "More formally, under sufficient smoothness conditions" - and what are these conditions?
> > >
> > > Please see equations 1.21, 1.22, and 1.23 in Karlin and Taylor textbook (as already cited in the paper with the corresponding pages for ease of access).
> > >
> > > > Why would limits (1) and (2) exist? Where do they come from? What is the reasoning? What's the deduction of their existence?
> > >
> > > These are standard assumptions for diffusion processes and they allow the integrals in the decomposition lemma to be well-defined (more formally, they are required for the Ito’s lemma). These assumptions will not hold for example if state and time are both continuous and there exists an *infinite* number of discontinuity in the state trajectory, or if for example there is an unbounded discontinuity in the state trajectory. There are several basic theorems for diffusions which assure such existence or to show that a process is diffusion. We do not present/repeat such basic results (as this is a conference paper, not a book chapter). We refer the reader to textbooks such as Karlin and Taylor.

---

> > > > ### Author Response · Authors · 2023-11-14
> > > > **Cont.**
> > > >
> > > > > Under process-based causality, I feel that several references are amiss. Yes those axioms (I and II) are mostly true (though (i) appears not to be within the space of quantum dynamics but let's not go there) but they would do well to be referenced. Hume argued the temporal priority principle a couple of hundred years ago, so this is nothing new and hence should be placed within the literature. Where does (iii) come from?
> > > >
> > > > Thanks for the pointer; we will clarify and add references. We did not mean for these axioms to be realised as our contribution (or being said for the first time), rather to summarize what is true and is required to be considered. David Lewis also talked about these (but not all in one place). As for the second point, you are right, in [pathological] cases like the edge of a blackhole or a time-machine, these axioms may be violated, but we do not consider such problems. Of note, (iii) is an axiom that we present to understand causation between events. Based on several practical cases of interest we argue that an event need not be necessary or sufficient to be identified as a cause of an event. For example, in many practical cases a patient might need both treatment A and treatment B to fully recover from disease X. Here neither treatment A or treatment B is in itself sufficient or necessary to recover from disease X, but are causes for the full recovery of the patient anyways.
> > > >
> > > > > You haven't specified what 'HP' stands for so 'HP settings' makes no sense.
> > > >
> > > > It stands for Halpern and Pearl. The explanatory sentence should unfortunately have been deleted in one of the last revisions. Will fix it.
> > > >
> > > > > "central idea behind Lewis definition" who is Lewis here? What's the reference to?
> > > >
> > > > David Lewis, the philosopher who invented the counterfactual view of causation (much before Pearl and others brought it into statistics). We add a sentence the first time Lewis is cited to clarify.
> > > >
> > > > >"causes, by themselves, increase the probability of their effects" that is nonsensical; are you arguing that if a cause exists then the effect may happen?
> > > >
> > > > Yes we are -- and this is sensical. In general, a cause only increases the probability of its effect and may not necessarily force the effect to happen. This is the *probabilistic* view of causation as put forward by Lewis and has also been adopted by many others (see Lewis, D. 1986. Postscript to ‘Causation’. In his Philosophical Papers, Vol. 2, pp. 175--84).
> > > >
> > > > Note that, (i) the deterministic view where “if a cause happens then effect will happen” is a special case of this view and fully consistent, and (ii) the probabilistic view still covers the possibility of a sufficient cause (one that forces the effect to happen).
> > > >
> > > > The reviewer can further refer to this work to better understand this fundamental philosophy of causation:  Hitchcock, Christopher, "Probabilistic Causation", The Stanford Encyclopedia of Philosophy (Spring 2021 Edition), Edward N. Zalta (ed.), URL = <https://plato.stanford.edu/archives/spr2021/entries/causation-probabilistic/>
> > > >
> > > > > "as the role of intervention is to mechanically separate the impact of a variable from the collective impact" what does that mean?
> > > >
> > > > We understand that several factors could be impacting the effect event of interest, eg. lung cancer. By the above phrase “as the role of intervention is to mechanically separate the impact of a variable from the collective impact” we mean that interventions like do(smoking=yes) or do(smoking=no) are made in order to understand the effect of smoking on lung cancer in isolation.
> > > >
> > > > > What is a "sticky value"?
> > > >
> > > > As explained in the paper right after this term is used: “once it is reached, grit will remain at one until B is forcefully reached, irrespective of any intrinsic or extrinsic future event.”
> > > >
> > > > > grammar
> > > >
> > > > We certainly proof-read the paper. We should highlight that we have not found severe grammatical errors as commented so strongly by the reviewer.
> > > >
> > > > > So far I do not following the reasoning in this section, specifically though why would this be true "This could prove useful in the context of causal discovery."?
> > > >
> > > > The definition of event allows for multiple state variables to shape an event. In *causal discovery* the $A$-event is not given and should be found. Using the individual contributions would help strip event $A$ to only include state components with genuine contribution (if $A$ is not selected wisely, then while $A$ as a whole may still be a cause, all its components may not necessarily contribute positively).

---

> > > > > ### Author Response · Authors · 2023-11-14
> > > > > **Cont.**
> > > > >
> > > > > >> “Modeling dynamical systems as SCMs is computationally and memory intensive to a prohibitive degree…” this is not true …
> > > > >
> > > > > >> if they are truly inadequate, as you claim, for dynamical settings then empirically show us why and show us how much better this theory is in comparison …
> > > > >
> > > > > First, the mentioned paper by the respected reviewer **has already been cited in our paper** (and we are certainly aware of many such methods).
> > > > >
> > > > > The Atari example comprises $4\times 84\times 84 = 28224$ state variables with 57 time-steps, summing up to more than **1.2 million variables**, each can have 256 discrete values. A causal graph with this number of nodes and **combinatorially larger** number of edges is “computationally and memory intensive to a prohibitive degree” as asserted in the paper. Also, note that this example only includes 57 time-steps, while a nominal game play can include thousands of time-steps (and normally much larger screen size than 84x84 pixels even after down-sampling), blowing up the graph far more immensely. Above all these, how do you want to intervene in an ongoing game with no access to the internal game engine to mechanically adjust state variables? Simply impossible.
> > > > >
> > > > > Similar arguments go for the other experiment in the paper, just as any other dynamical problem with many state variables and a long run-time.
> > > > >
> > > > > **We do maintain the validity of our statement** in the paper that other methods in the existing literature are not even possible to be used as baselines.
> > > > >
> > > > > As for the second part of the question, the answer is already given in the paper: our method fully accurately marked the causes with no access to internal models or human knowledge.
> > > > >
> > > > > > caption of the figures
> > > > >
> > > > > We will add more explanation (as space allows). There is also a dedicated section in the appendix which details each experiment (we add more details there too). In addition, we will release the complete code to reproduce all the presented results.
> > > > >
> > > > >  > … but I am struggling to understand how this is at all more useful?
> > > > >
> > > > > Honestly, this comment surprises us the most. The cause of losing the ball has been identified by showing the exact pixels (i.e., the state variables, which shape event $A$) at the correct time-step, solely using the observational (pixel-level) data!! This directly showcases how our framework can identify causes of a given event **with NO human-level knowledge,** most certainly **no graph,** and purely based on **data.** This exemplifies numerous dynamical systems, which can benefit from the very same method. You can think of identifying the causes of why a given patient has experienced a stroke in the ICU at a certain time, or why some section of a nuclear reactor has exploded, or why a certain protein has stopped developing, or why a chemical plant has failed to shut down, … (and the list goes on).

---

> > > > > > ### Comment · Reviewer_6KUF · 2023-11-22
> > > > > > **Acknowledging**
> > > > > >
> > > > > > Thanks to the authors for a thorough response. I confirm that I have read their review and have no further questions or comments.
> > > > > >
> > > > > > Thanks also for pointing out my my misunderstanding and providing clarification.

---

### Official Review · Reviewer_7WTp · 2023-10-31

**Soundness:** 3 good
**Presentation:** 3 good
**Contribution:** 3 good
**Rating:** 6
**Confidence:** 3

**Summary:**

The paper proposes a new framework in a stochastic process setting. In contrast to previous work on continuous-time causality which aims to compute the effect of certain prespecified interventions (in SCM-type settings), the paper seeks to answer questions of the type "Given an event B, can we say whether another event A was a cause of B?". A central idea is the one of grit: If A is the cause of B, the corresponding grit (minimum probability of occurrence of B under all possible policies) should increase. Based on this intuition, the paper provides a rigorous definition of the cause of a given event, as well as the computational tools to verify this. Finally, the benefits of the proposed framework are shown in two experiments.

**Strengths:**

- The paper deals with an important and often overlooked question. Most works on causality deal with obtaining the effect of prespecified causes (interventions). There is not much work on the reversed problem, i.e., reasoning about causes given an effect/event.
- The proposed dynamic framework is novel, intuitive, and theoretically sound. To the best of my knowledge, the paper is the first to propose a framework for "reverse causality" in a dynamic setting.
- The paper is well-written and the main ideas are easy to grasp, even though the topic is quite abstract/conceptual. The authors first provide the necessary intuition before formalizing the ideas.
- Promising experimental results

**Weaknesses:**

- The general problem setup reminds me of what some in the field call "reverse causality". I.e., the aim is not to obtain the effects of causes (interventions), but rather obtain the causes of effects. I am not an expert in this area, but I think there are some existing works on this that are not mentioned in the paper, e.g., Gelman and Imbens (2013). I do not think that these approaches build upon a dynamic setting, but I still think that the paper would benefit from a section that relates these approaches.
- Definition of cause/grit. I am wondering whether an event $A$ could be a cause of an event $B$ even though the expected grit of $B$ does not increase from before to after $A$. Imagine a medical setting, where a doctor prescribes treatment (event $A$), that only works in combination with another treatment (event $C$) to cure the patient (event ($B$). ($A$ -> $C$ -> $B$). Then $C$ would be a cause but $A$ would not be a cause because the grit would not increase from before to after $A$ (since the grit is the minimum probability of $B$ over all possible policies, also the one that does not administer $C$). However, intuitively both $A$ and $C$ should be causes.
- No code is provided (this is minor as the contributions of the paper are mostly theoretical)

**Questions:**

- Why is the notion of grit the correct way of defining causes (see weaknesses)? I am open to increasing my score if the authors make a convincing case.
- The notation of the state $X(t)$ seems inconsistent. Should it not depend on the actions $u(t)$? E.g., in Eq.(1) and (2) $u$ only appears on the right-hand side of the equations. Furthermore, $V^\pi$ on page 3 does not depend on $\pi$.

---

> ### Author Response · Authors · 2023-11-14
>
> We appreciate the reviewer for the positive and encouraging feedback. We discuss the questions below. Please do let us know if the explanations are clear and whether you have more questions.
>
> > The general problem setup reminds me of what some in the field call "reverse causality".
>
> Thanks for the pointer, we would add the citation. This seems to be a short paper that suggests forward and reverse questions. While not providing any actual formulation (and computational framework), the view is interesting and must be acknowledged.
>
> > Definition of cause/grit (medical event A -> C -> B).
>
> In your example, if $C$ happens (or can happen) with some non-zero probability, even unknown or non-stationary, then $A$ is a cause of $B$ and is also correctly captured by our theory [the grit indeed increases by $A$ as shown below]. On the other hand, if $C$ is chosen not to happen at all, then $B$ will never happen. In this case, $A$ will not be considered a cause because we violate condition C1 (in our definition of causation), where $A$ must happen before $B$ - here $B$ never happens! We note that in the lack of event $B$, there is no causal question. Put it a bit differently, $A$ is a cause of $B$ conditioned on $C$ being selected with non-zero probability (and this is exactly what our framework deduces). That is, grit is meaningful only if $B$ is not unlikely to happen.
>
> As a side note, we have also observed this computationally in the Diabetes example. W.r.t. insulin intakes, we observed that **any** insulin intake shows an increase in expected grit -- even when it is insufficient to cause event $B$ by itself and at least another sufficiently large intake is required. This is precisely what is expected from the theoretical analysis.
>
> **Analysis:** First, please recall that the definition of causation asserts the **expected** change of grit must be positive from before to after $A$. Let $C$ happen with probability $p > 0$ and if it happens, the grit of $B$ increases by $\alpha > 0$ (since $A$ has already happened). Then $\mathbb{E}\Delta\Gamma_{B}(A) = p\times\alpha + (1-p) \times 0 > 0 $. Hence, grit of $B$ cannot be the same from before to after $A$, and indeed **it increases**. Thus, $A$ is also a cause unless either $p$ or $\alpha$ is zero.
>
> > Why is the notion of grit the correct way
>
> As we discussed above, the notion of grit works fully inline with what is expected intuitively as a causal relationship (even when it looks counterintuitive at the first glance). The minimum probability provides the natural way of dealing with possible future actions, as it ensures that $A$ is named a cause of $B$ only if it is so under any (even pessimistic) future trajectory. Importantly, if there is no action, then it automatically retrieves the standard definition (that $A$ should increase *probability* of $B$, rather than *grit* of $B$). Moreover, if a fixed policy is chosen to follow, either in general or only after event $A$, our framework still works readily, as $V^*$ will automatically be replaced by $V^{\pi}$.
>
> > No code is provided
>
> We will release the complete code that reproduces all the results presented in the paper (and can easily be modified for other use-cases).
>
> > The notation of state $X(t)$ …
>
> This is a common way, which basically only sets time as the base argument (mostly to parallel the discrete-time format). It’s being used in literature, see for example RL/DP papers in continuous time, such as this one:
>
> Kim and Yang, ‘Hamilton-Jacobi-Bellman Equations for Q-Learning in Continuous Time’, in Conference on Learning for Dynamics and Control, 2020, vol. 120, pp. 739–748.
>
> > notation $V^{\pi}$
>
> We fixed it; thanks for the pointer. $\pi$ needs to be in the condition of expectation.

---

### Official Review · Reviewer_nNBo · 2023-11-05

**Soundness:** 2 fair
**Presentation:** 1 poor
**Contribution:** 2 fair
**Rating:** 3
**Confidence:** 4

**Summary:**

The paper introduces a formalism to talk about causality in dynamic settings, and cast this as a reinforcement learning problem. The paper studies the case when the effect is given, and one has to reason about plausible causes.

**Strengths:**

- Connecting grit (“minimum probability that event happens given a current state”) and reachability (“ maximum probability of an event occurrence starting from given state”) to optimal value functions corresponding to constructed reward functions is an interesting idea.
- The proposed method can operate on raw observations without assumptions about system equations and causal graph.

**Weaknesses:**

- The reviewer find the paper very hard to read and understand.
- Some of the assumptions are already well known in causality (“Causality necessitates time”. “ Cause happens before effect”).
- The reliance of the proposed method on optimal value function corresponding to reward function seems a strong one.
- will be interesting to investigate how the proposed method works as the function of number of variables.

**Questions:**

Refer to Weaknesses

---

> ### Author Response · Authors · 2023-11-14
>
> We thank the reviewer for the feedback. We address each question/concern bellow. Please do let us know if these discussions are clear and satisfactory and whether you have more questions.
>
> > The reviewer find the paper very hard to read and understand.
>
> We would like to highlight that this is a deeply interdisciplinary subject by nature and a certain level of complexity is unavoidable. This paper covers materials at the intersection of different communities (causality, systems control, reinforcement learning, and even philosophy). We tried to keep the presentation more formal and less descriptive with the hope that it helps the reader to follow the ideas in a precise manner, regardless of differences in community-related aspects. We indeed wrote several drafts of this work, and based on various internal feedback; the current version seems to be the most reliable as read by people with different backgrounds.
>
> We, however, will do modifications before final submission to improve the presentation as much as possible.
>
>
> > Some of the assumptions are already well known in causality (“Causality necessitates time”. “ Cause happens before effect”).
>
> We are respectfully not sure why this point is set as a weakness?
>
> In any case, we would like to emphasize that these are NOT our assumptions, rather these are basic axioms which are well-known and are true in any causal arguments. We simply place them at the forefront. We will modify the text to clarify this point.
>
> > The reliance of the proposed method on optimal value function corresponding to reward function seems a strong one.
>
> It is a fair point but it is also the case for **any** method that uses neural nets or approximation techniques (and we add a statement to reflect on this in the paper). Nonetheless, what is important here is that, using optimal value functions, our framework provides a strong basis (with formal grounds) to design a learning problem and gives all the computational tools to establish causal links in dynamical systems -- something which did not exist before, at least not at this level of formal rigour and explicitness, while still practically applicable and useful. Then, in practice, the optimal values will be replaced with the *learned* ones from data. As with ANY learning problem, the results will be reliable to the extent of approximation errors; this is not a specific problem of our framework, rather is a generic issue.
>
> Of note, additional theoretical arguments can also be made here. For example, if the approximated value function is off by some maximum amount, then under certain assumptions, the degrade in validity of causal arguments will also be bounded (proof sketch: the bound over values will map to the same bound over grit, which will then be used in the decomposition lemma to bound the error in each component). Or, if the value of a certain policy is used in place of $V^{*}$, the causal arguments will be completely valid if that policy is followed after event $A$ (this result is immediate from the definition of value functions).
>
> We would like to remark that presenting such results can go much beyond the scope of a single conference-paper.
>
> > will be interesting to investigate how the proposed method works as the function of number of variables.
>
> Our framework scales well to even huge state-spaces simply because it maps the problem into RL settings (and current deep RL methods can easily handle extremely large state-spaces). Even in the provided Atari problem, the state comprises 28224 variables (corresponding to the pixels of four consecutive game screens)! The T1 diabetes simulator, on the other hand, deals with 13 state variables and 2 action variables. This reflects the range of variables our proposed method can handle.
>
> In a related context, as the base problem here is an RL problem (due to our Value Lemma), any value-based RL technique can be used, and the scalability will be inherited from the properties of the RL algorithm.

---

### Official Review · Reviewer_f34o · 2023-11-05

**Soundness:** 4 excellent
**Presentation:** 2 fair
**Contribution:** 3 good
**Rating:** 8
**Confidence:** 3

**Summary:**

This paper presents a novel dynamical systems-based perspective on discovery of the causes for a given effect. They extend Lewis [1979, 2000] notion of a cause as an event that precedes the effect event and increases the effect's probability relative to a counterfactual world where the cause did not occur. The extension to MDPs takes the min over all actions that lead to the event. Given this notion of causality, they show that the key quantities, grit (the minimum probability that an event occurs from a given state), and reachability (the analogous maximum probability), can be computed by solving an appropriately designed reinforcement learning problem. This makes the framework generically applicable which they demonstrate with experiments from Atari's Pong (with pixel-level observations), and a diabetes simulation.

**Strengths:**

This is ambitious and interesting work, so much so that I was somewhat surprised to see it submitted to ICLR which is probably not the best venue to debate the merits on novel views on causality (one would ideally want input from philosophy). I didn't love the presentation (see weaknesses below), but I think that the method that they arrive at is interesting and relatively practical.

**Weaknesses:**

My main critique is around the structure of the paper. It would be so much easier to follow if all the critiques of the existing frameworks in causality were made with reference to one of the examples that you use in the experiments. As written, we have to take it on faith (or go and read the cited references) that the current frameworks are insufficient for what you're trying to do, and it is only when you get to the experiments that it becomes a little clear why, e.g. a pixel level observation would make the Pearlian framework difficult to apply, or why interventions would be tricky in the diabetes simulator. And perhaps most importantly, it would help explain the need to map the estimation of grit and reachability as an RL problem. If it was up to me, I this paper in the reverse order of how it is currently written: first set up a problem from the experiments, then explain the key queries that you would like to answer from section 4, and then introduce how they can be answered by formulating this as an RL problem (section 3).

MDP:
 - when the MDP is introduced, it is not clear how it relates to the system of interest - was only later that I realized that the MDP is used for estimation of grit and reachability, and not part of the definition of causality.

Minor:
 - top of page 3: "straight" -> "straightforward"
 - "In the HP settings of causation" - what is HP here?

**Questions:**

Does this framework need the ability to take actions (i.e. do we need an MDP), or is that just an estimation technique? I think it is the former, but  that implies that you have a simulator that effectively plays the role of the causal model in that you can get at counterfactual queries by trying different actions from the same state. If this is the case, I would like it made explicit in the limitations. E.g. if I just observe a dynamical system but don't have the ability to take actions, I can't infer causality in this framework, right?

---

> ### Author Response · Authors · 2023-11-14
>
> We thank the reviewer for the very encouraging feedback. Here we address the questions; please let us know if there are further questions/clarifications.
>
> > structure of the paper
>
> We do appreciate the suggestion (and indeed liked it). However, the page limit does not allow for such a presentation, unless a significant portion is omitted. As this paper covers materials at the intersection of different core topics (causality, systems control, reinforcement learning, and even philosophy), we believe it would be required to cover them all and there is little room to shorten the paper. We tried to keep the presentation more formal and less descriptive/intuitive with the hope that it helps the reader to follow the ideas in a precise manner, rather than motivating at the forefront. We indeed wrote several drafts of this work, and based on various internal feedback; the current version seems to be the most reliable as read by people with different backgrounds. We, however, will do modifications before final submission to improve the presentation as much as possible.
>
> > what is HP
>
> It stands for Halpern and Pearl. The explanatory sentence should unfortunately have been deleted in one of the last revisions. Will fix it.
>
> > MDP
>
> We will add a clarification why MDP is defined (when it is introduced) to help address your point.
>
> > Does this framework need the ability to take actions (i.e. do we need an MDP), or is that just an estimation technique?
>
> Our framework does not *need* the ability to take actions, and you certainly can use this theory for a system with only states (no action).
>
> The reason to use MDP is basically the other way round: a system/environment may *include* actions and the framework needs to be able to cover such cases as well; hence, MDP is needed in general. Moreover, as we discussed in the paper, an event may also be defined as taking an action (a change in an action variable) rather than a change in a state variable. Our theory covers such cases too, which is again why we need MDP in general.
>
>  > I just observe a dynamical system but don't have the ability to take actions, I can't infer causality in this framework, right?
>
> You definitely can. This case can happen if either there is no action in the system by nature, or a fixed policy is taking care of actions with no freedom (hence, there is effectively no action). In all state-only problems, you simply use $\Gamma_{B}(X)$ and $V_{\Gamma}(X)$ and the rest of the theory remains valid and intact.

---

### Meta-Review · Area_Chair_NyAn · 2023-12-09

**Metareview:**

The paper introduces a novel dynamical systems-based perspective on causality, extending Lewis's notion of cause and exploring the concepts of grit and reachability in the context of reinforcement learning. The proposed framework is demonstrated through experiments on Atari's Pong and a diabetes simulation. The reviewers generally find the work ambitious and interesting, acknowledging its theoretical soundness and practical relevance. However, they raise concerns about the paper's presentation, the structure of the arguments, and the clarity of certain concepts.

Addressing these concerns, providing a clearer connection between existing frameworks and the proposed approach, and improving the explanation of experimental results could enhance the paper's overall quality. The novelty and significance of the proposed framework are acknowledged, and addressing the reviewers' feedback could increase the quality of this paper.

**Justification For Why Not Higher Score:**

The current presentation needs significant improvement.

**Justification For Why Not Lower Score:**

The novelty and significance of the proposed framework are acknowledged.

---

### Decision · Program_Chairs · 2024-01-16

Accept (poster)